# Early-phase impact of obesity-associated stress on murine vascular smooth muscle cells depends on EGFR and sex

Virginie Dubourg [1,2], Sindy Rabe[1,2], Narjes Nasiri-Ansari[1], Michael Kopf[1], Sigrid Mildenberger[1], Gerald Schwerdt[1], Barbara Schreier [1,3] & Michael Gekle [1,3] ✉

Obesity leads to vascular dysfunction mediated partially by the vascular smooth muscle cell (VSMC) EGF-receptor (EGFR). We investigate the impact of obesity-associated metabolic and humoral stress on primary murine VSMC with conditional EGFR knockout (KO) and wildtype (WT) VSMC, focusing on early-phase impact to test the hypothesis of an EGFR-dependent stressor synergism. Cells are exposed to three stress conditions (high glucose + free fatty acids; angiotensinII + noradrenaline; combined = all stressors) and bulk RNA-sequencing with bioinformatics analysis, followed by phenotypical assessment is performed. RNASeq-results show stressor synergy in male WT-VSMC but not in KO VSMC or endothelial cells (EC). Bioinformatic analysis predicts dysregulation of functions related to DNA-synthesis/cell cycle, lipid handling, contraction and motility for male WT-VSMC. Functional validation confirms synergy concerning DNA-synthesis and lipid accumulation in male WT-VSMC but not in female WT-VSMC. Altered contraction or motility are not confirmed. Male WT-VSMC show higher EGFR-expression than female WT-VSMC and respond with enhanced SRF$^{S103}$-phosphorylation, a classical downstream target of EGFR, to the stressors. Obesity-associated metabolic and humoral stressors induce synergistic transcriptomic effects in male WT-VSMC, initiating proliferative and lipogenic dedifferentiation. This early-phase effect requires EGFR and was not observed in female VSMC.

Obesity and diabetes mellitus type 2 (T2DM) are a global health concern, with prevalence rising[1]. T2DM significantly impacts vascular smooth muscle cells (VSMCs), contributing to increased cardiovascular risk: VSMCs from T2DM patients exhibit phenotypic alterations, including enhanced proliferation, adhesion, migration[2] and display impaired function compared to non-diabetic controls[3]. This altered VSMC phenotype contributes to vascular complications and intimal hyperplasia, potentially leading to end-organ damage, e.g. renal failure. Understanding these mechanisms is crucial for developing targeted therapies to address vascular complications in T2DM patients.

The plasticity of VSMC, which can switch between contractile and various transdifferentiated states, plays a crucial role in vascular homeostasis and remodeling[4]. Furthermore, endothelial dysfunction, characterized by reduced nitric oxide bioavailability, impaired vasodilation, increased vascular tone, and a prothrombotic state contributes to functional and structural remodeling in diabetic patients[5]. Given the limited effect of blood glucose control on T2DM-associated cardiovascular risk,

research has been focusing on alternative therapeutic strategies, such as targeting VSMC[4]. The metabolic milieu of T2DM, which includes insulin resistance, hyperglycemia, elevated plasma free fatty acids, overactive renin-angiotensin system and elevated sympathetic tone[6,7], is supposed to contribute to vascular dysfunction through various mechanisms in different cell types. Notably, vascular dysfunction can occur already in prediabetic states, suggesting a continuum of cardiovascular risks as metabolism worsens[8].

EGFR (ERBB1) is activated by epidermal growth factor (EGF) or heparin bound EGF (HB-EGF), modulating cell differentiation, migration and matrix homeostasis[9]. EGFR can also be transactivated, thereby contributing to cardiovascular dysfunction and remodeling[10,11]. Recently, the relevance of VSMC-EGFR for structural and functional vascular remodeling as well as for subsequent renal end organ damage was shown[12-14]. It has been suggested that VSMC-EGFR contributes to obesity- and T2DM-associated vascular alterations[11,15-19], and that hyperglycemia is correlated with an enhanced vascular EGFR activity. There is also evidence for an

[1]Julius-Bernstein-Institute of Physiology, Martin Luther University Halle-Wittenberg, Halle, Germany. [2]These authors contributed equally: Virginie Dubourg, Sindy Rabe. [3]These authors jointly supervised this work: Barbara Schreier and Michael Gekle. ✉e-mail: michael.gekle@medizin.uni-halle.de

impact of free fatty acids on EGFR[20–22]. In addition, systemic EGFR-kinase inhibitors improved vascular function in diabetic animals[16–19,23–25].

In recent studies we showed that deletion of VSMC-EGFR mitigated T2DM/obesity-induced vascular functional, structural and transcriptome alterations drastically in vivo[26], using a knock out mouse model with inducible deletion of the EGFR in VSMC[12,27]. These data strongly suggested VSMC-EGFR as prerequisite for HFD-induced transcriptome alterations followed by vascular remodeling. Furthermore, the protective effect of VSMC-EGFR-KO on vascular remodeling led to partial prevention of renal end-organ damage, normalizing glomerular function and reducing tubule-interstitial dyshomeostasis as well as the development of albuminuria.

At the cellular level transcriptional actions of EGFR were potentiated by glucose and pathway analysis identified the serum response factor (SRF) as an activated vascular transcription regulator during T2DM/obesity. It was concluded that VSMC-EGFR is required for comprehensive T2DM/obesity-induced functional vascular remodeling, endothelial dysfunction and renal end-organ damage. By contrast, the endothelial cell (EC)-EGFR has been shown to be of minor importance for basal vascular and renal function, as well as for T2DM/obesity-induced functional vascular remodeling, endothelial dysfunction and renal end-organ damage[28].

These in vivo studies represented early phases of T2DM/obesity with mild vascular dysfunction and mild end-organ (kidney) damage. They also confirmed the systemic relevance of VSMC-EGFR for vascular pathophysiology during obesity with T2DM, already in the onset phase of the disease. But these studies do not allow to evaluate a direct impact of T2DM/obesity-associated stress factors on VSMC and EC.

The objective of our study is to investigate the direct early-phase effects of T2DM/obesity-associated stressors on vascular cells in order to gain more knowledge regarding pathophysiological events occurring before major phenotypic cellular changes and vascular structural alterations develop. Specifically, our study investigates the hypothesis that a stressor synergism acts on VSMC in an EGFR-dependent manner. For this purpose, we used ex vivo murine primary vascular cells ex vivo that we subjected to T2DM/obesity-associated metabolic (high glucose and free fatty acids) or/and humoral (angiotensin II and noradrenaline) stressors. Furthermore, we addressed the role of VSMC-EGFR for these early-phase events using EGFR-KO VSMC from a genetic mouse model. Better understanding of these initial events may support the development of preventive strategies.

## Results
### Cell viability
First, we investigated whether the humoral stressors (angiotensin II + norepinephrine; AII + NE), the metabolic stressors (high glucose concentration + free fatty acids; HG + FFA) or their combination (ALL) affected cell viability. For this purpose, the following parameters were determined: cell (nuclei) count, protein (per cell and per area), caspase-3 activity (as a marker for apoptosis) and LDH-release (as a marker for necrosis). Supplementary Fig. SF2 shows that the stressors did not affect cell viability.

### Gene expression analysis
Bulk RNA-Seq was performed with samples from male animals. The variability of gene expression throughout the samples was assessed using the average relative standard deviation (with the relative standard deviation calculated as standard deviation(FPM)/mean(FPM) for each gene). It was similar for WT VSMC and KO VSMC and all four incubation conditions (Supplementary Fig. SF3). Thus, we could apply one threshold (calculated as 3 × average relative standard deviation) for the parameter fold-change ($|\log_2FC| > 0.64$).

First, we analyzed the RNA-Seq data in a directed approach, i.e. we investigated potential changes in RNA expression for genes known to be typically affected during pathological vascular alterations (biomarker genes)[29,30]. Because our experimental design provided strictly connected sets of samples for control, metabolic stressors, humoral stressors and combined stressors originating from the same animal, at the same cell passage, treated

at exactly the same time, it allowed us to calculate the relative effects of the stressors in a paired way (stressor effect = $expression_{stressors}$/$expression_{control}$, with the expression levels corresponding to the FPM obtained for each corresponding set of paired samples). Subsequently, we calculated the 99% confidence intervals and tested the exclusion of the value = 1 (corresponding to a difference from controls with $\alpha < 0.01$). Supplementary Fig. SF4 and Supplementary Data 01 show the list of genes investigated and the effects of the stressors on their expression. The data show that only few of the biomarker genes were affected by all stressors in WT VSMC and the changes were not indicative of phenotypic switching. This concurs with our purpose to investigate early-phase effects. The changes in EGFR-KO VSMC were even smaller (only 5 genes were affected compared to 16 in WT VSMC), already indicating a possible contribution of EGFR. The effect in EC was also minor (5 genes affected).

### Unbiased gene expression analysis
Next, we performed an unbiased gene expression analysis including all protein coding and lncRNA coding genes. Applying the $|\log_2FC|$ threshold of 0.64, together with the double-significance criteria (i.e. differential expression DE indicated by DESeq2 AND edgeR) and an FPM threshold ( > 3 FPM) resulted in the detection of quantitative changes in RNA expression as shown in Fig. 1.

In WT VSMC the individual effect of HG + FFA was stronger than that of AII + NA (Fig. 1A) and the number of differentially expressed genes (DEG) in the presence of all stressors was much higher than the sum of genes regulated by HG + FFA or AII + NA (Fig. 1A). Thus, there is a quantitative over-additive (synergistic) action of the stressors concerning alterations of the transcriptome. The number of up-regulated genes was slightly higher, but still in a similar range, as the number of down-regulated genes for all conditions (Fig. 1A). In KO VSMC, this quantitative synergistic effect was completely absent (Fig. 1B) and AII + NA exerted no effect, whereas the effect of HG + FFA was quantitatively comparable to WT VSMC. Thus, the quantitative synergism of HG + FFA and AII + NA as well as the effect of AII + NA require EGFR expression, whereas the quantitative effect of HG + FFA is largely independent of EGFR expression.

The scatter plots in Fig. 1A, B show the comparative analysis of $\log_2FC$ values in the presence of either ALL stressors or HG + FFA for the group of DEG induced by ALL stressors in WT VSMC and KO VSMC. The scatter plot in Fig. 1A, with a slope of 0.497 (and therefore much lower than 1), shows that most of the genes affected by ALL stressors are indeed not affected by HG + FFA in WT VSMC. The scatter plot in Fig. 1B, with a slope of 0.87, shows that in KO VSMC most of the DEG are affected by HG + FFA and ALL stressors in a similar way.

Supplementary Figs. SF5A–C show Venn diagrams of DEG overlaps and lists with effect sizes and confidence intervals. The effect sizes elicited by the stressors and their confidence intervals were determined from the impact of the two stressor conditions in a paired manner (see Methods) relative to the respective controls of each experimental set, to rule out that genes were not included in one of the two lists due to the high stringency conditions (DESeq2 AND edgeR positive) applied. For KO VSMC the Venn diagram in SF05A show groups of DEG apparently affected by HG + FFA or ALL stressors only. However, the paired analyses (scatter plot in Fig. 1B with a slope of 0.87; lists II and III with the confidence interval limits in SF5A) show that most of the DEG are affected by HG + FAA and ALL stressors in a similar way in KO VSMC. Thus, KO VSMC respond to HG + FFA but not to AII + NA, also when ALL stressors are present.

The results from these qualitative comparisons confirm the conclusion of an overadditive (synergistic) effect of HG + FFA and AII + NA in WT VSMC but not in KO VSMC. Furthermore, the list I in SF5A shows that all DEG affected by HG + FFA are also affected by ALL stressors in WT VSMC. Figure 1D shows aggregate mesh plots, generated by Sigma plot 14, of the $\log_2FC$ for all three stressor conditions for all protein coding RNA and lncRNA of WT VSMC and compares the measured values for ALL stressors with the calculated additive ones.

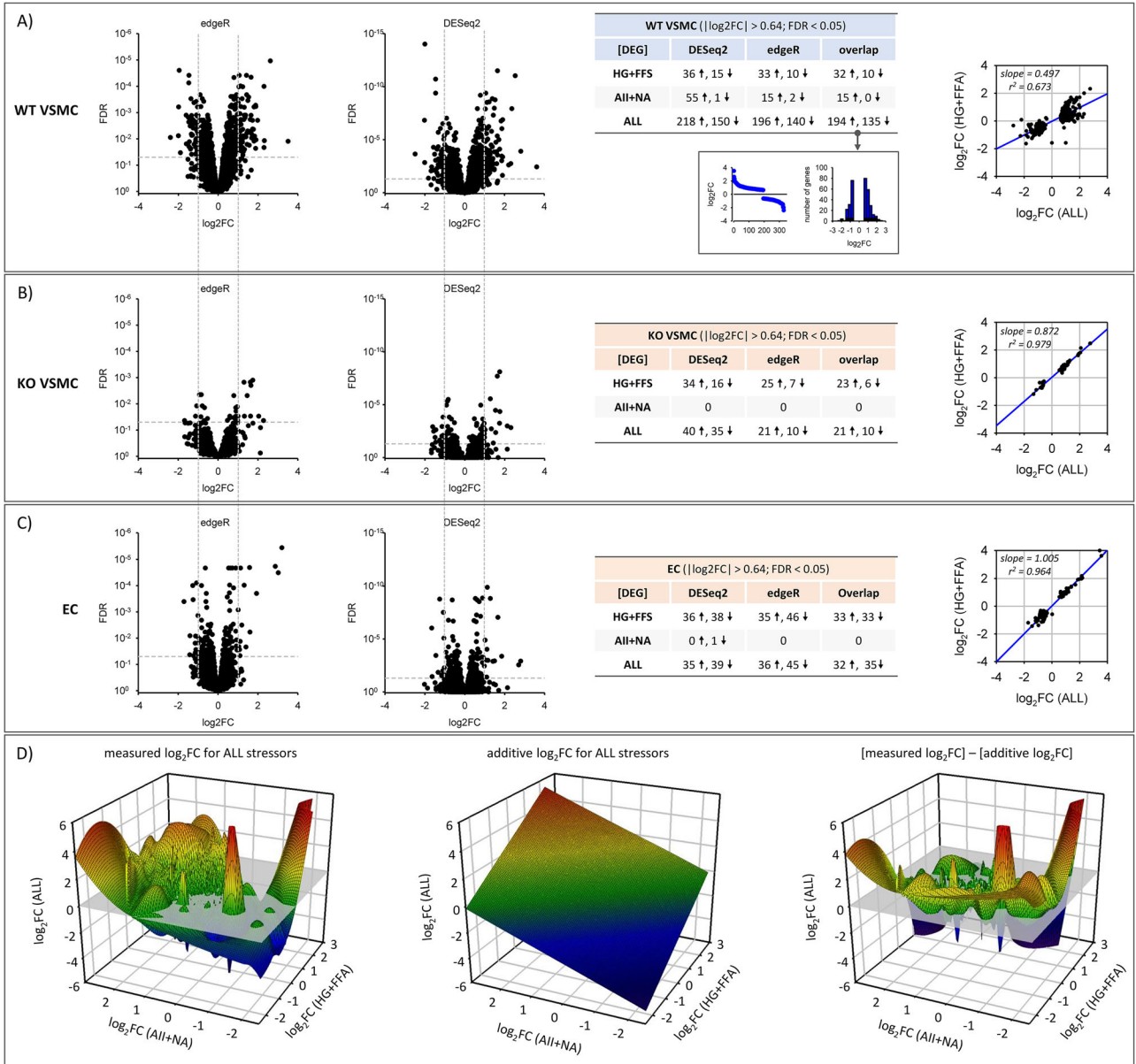

**Fig. 1 | Differential gene expression analysis.** Number of differentially expressed genes (DEG) after exposure to metabolic stressors (high glucose + free fatty acids, HG + FFA; humoral stressors (angiotensin II + noradrenaline; AII + NA) or the combination of both (ALL) in male wildtype VSMC (WT VSMC, **A**), EGFR-KO VSMC (KO VSMC, **B**) or endothelial cells (EC, **C**). For VSMC and each condition 4 independent biological replicates were included in the differential expression analysis, for EC 10 samples were included ( = total of 72 samples). The left panels show volcano plots for false discovery rates (FDR) and effect size (log2FC) determined either by edgeR or DESeq2 for cells exposed to all stressors. The tables to the right show the number of up (↑) or down (↓) regulated DEG after filtering for the log2FC and FDR thresholds. All FDR-, log2FC- and FPM-values for all three cell types are presented in Supplementary Data 16 and 17. **D** Three-dimensional mesh plot, generated by Sigma plot 14, combining the log2FC data for all protein coding RNA and lncRNA of WT VSMC for all three stressor conditions. The first plot shows the measured log2FC values for ALL stressors, the second plot the additive log2FC values (humoral effect + metabolic effect) and the third plot the difference of measured and additive values.

In addition, we compared the DEG induced by HG + FFA in WT VSMC and KO VSMC (Supplementary Fig. SF5B). Again, the Venn diagram shows DEG apparently affected by HG + FFA in WT VSMC or KO VSMC only. Yet, the paired analysis shows that most of the DEG are affected by HG + FFA in WT VSMC and KO VSMC in a similar way (see slope of the scatter plot = 0.918 and DEG lists with CI limits). Supplementary Fig. SF5C shows the FPM values for the 15 genes affected by AII + NA in WT VSMC. These genes were not affected in KO VSMC.

The mean variability of gene expression in EC was higher compared to the one in VSMC but similar in all four incubation conditions (Supplementary Fig. SF3), resulting in a calculated threshold of 1.0 for the parameter

fold-change. Supplementary Fig. SF6 shows the genes affected by ALL stressors in EC, after applying the log2FC threshold of 1.0. However, to make the results more comparable to those from VSMC, we also performed the analysis with the threshold |log2FC | > 0.64. The results obtained for this log2FC threshold, together with the double-significance criteria (i.e. differential expression DE indicated by DESeq2 AND edgeR) and an FPM threshold ( > 3 FPM) resulted in quantitative changes of RNA abundance as shown in Fig. 1C. In contrast to WT VSMC there was no synergism of metabolic and humoral stressors (see slope of the scatter plot in Fig. 1C = 1.01 and DEG lists IV and V in SF5A). Furthermore, humoral stressors led to no change and the quantitative effect of metabolic stressors

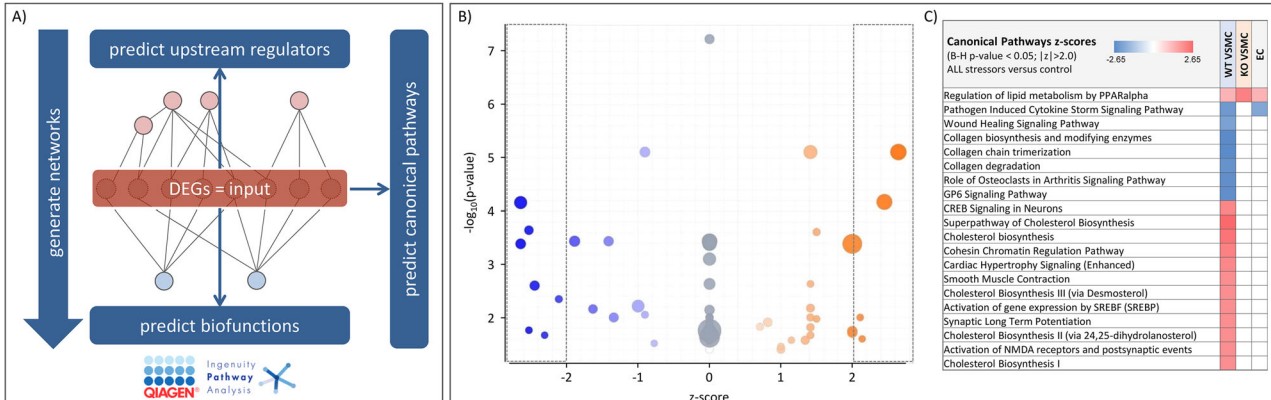

**Fig. 2 | Pathway and function enrichment analysis by IPA. A** Analysis scheme. The log₂FC-weighted lists of DEG served as input for the analysis to predict involved canonical pathways, upstream regulators and biofunctions as well as the direction and extend of change (z-score; activation or inhibition). **B** IPA results for canonical pathways using DEG of WT VSMC induced by exposure to all stressors. **C** The results for the canonical pathways predicted to be regulated by all stressors in at least one cell type were aligned. Presented pathways were filtered for z-score > 2 or < −2 and B-H corrected *p* value < 0.05. The complete data sets are shown in Supplementary Data 02–05. The genes used for the different enrichment analyses are presented in Supplementary Data 18–23.

was comparable to the one in VSMC. Thus, EC behaved similar to EGFR-KO VSMC.

Next, we performed functional enrichment analysis on the lists of regulated genes (results of the differential expression analyses) combining pathway analysis with the Ingenuity Pathway Analysis (Fig. 2A, IPA, Qiagen, Hilden, Germany) software (including Canonical Pathways, Upstream Regulator and Downstream Effects Analyses; that are features not included in g:Profiler) with gene ontology enrichment analysis with g:Profiler (http://biit.cs.ut.ee/gprofiler/;[31]) and protein interaction networks with STRING 12.0[32].

Figure 2 shows the IPA analysis results for canonical pathways (CP) of WT VSMC (Fig. 2B, C) and KO VSMC (Fig. 2C) as well as EC (Fig. 2C) exposed to all stressors. Exposure of WT VSMC to all stressors simultaneously is predicted an activation of lipid synthesis and a dysregulation of collagen homeostasis with no clear direction, because collagen biosynthesis and degradation are supposed to be reduced. For KO VSMC only the CP "regulation of lipid metabolism by PPARalpha" was predicted (-log₁₀(B-H *p* value) = 4.36; z-score = 2.24). The complete data sets are shown in Supplementary Data 02–05.

The results of the analysis for diseases and biofunctions (predictions of events downstream of the measured gene expression regulation) are shown in Fig. 3 and Supplementary Data 06–09. Exposure of WT VSMC to all stressors simultaneously is predicted to lead to enhanced proliferation and contraction of cells. "Smooth muscle contraction" is also a predicted CP (Fig. 2).

The results of upstream regulator analysis (URA - predictions of events upstream of the measured changes in gene expression) for WT VSMC exposed to all stressors simultaneously are shown in Fig. 4 and Supplementary Data 10–13. The fact that glucose and free fatty acids were identified is not surprising but can be taken as an indicator of IPA prediction fidelity. In the group of URA that may act from outside the cell or transduce outside signals amphiregulin (AREG), hepatocyte growth factor (HGF), receptor tyrosine kinases ERBB2 and ERBB3 are predicted. Together with regulator the NRAS they form a small cluster, indicative of signaling via the EGFR family.

Finally, Supplementary Fig. SF7 shows regulator effects networks (RE) set up by IPA, based on the results obtained for WT VSMC exposed to all stressors. These networks link the considered regulated genes with predicted upstream regulators and associated downstream events. The ones shown in SF7, which were filtered for high consistency scores (measure of the coherence of the links between the different levels of the networks and the database annotation), all converge on cell proliferation, or in other words, to cell cycle.

Results of gene ontology enrichment analysis with g:Profiler also comprised terms related to proliferation, cholesterol biosynthesis and extracellular matrix (Supplementary Data 14 and 15). Protein network analysis of predicted associations by STRING confirmed the predictions of the IPA analysis (Supplementary Fig. SF8), identifying clusters a.o. for cell proliferation, vessel contraction and sterol biosynthesis. On the other hand, the KO of EGFR in VSMC clearly prevented the dysregulation of pathways and functions according to the IPA analysis (Figs. 2–4).

Supplementary Fig. SF9 shows the results of IPA analysis on DE genes during exposure to all stressors in EC, which predict in principle only the expected cellular reaction to metabolic stressors and no effect of humoral stressors (IPA analysis with DE genes during exposure to metabolic stressors gave similar results). Thus, the response of EC to all stressors is similar to the one of EGFR-KO VSMC.

Supplementary Fig. SF10 shows the comparison of the three cell types (WT VSMC, EGFR-KO VSMC and EC) exposed either to all stressors or to metabolic stressors for canonical pathways, disease and biofunctions and upstream regulators. WT VSMC exposed to all stressors differ clearly from the other 5 groups.

## Cell cycle, proliferation

First, we compared nuclear DNA synthesis rate (BrdU-incorporation) and the relative cell cycle distribution of WT and EGFR-KO VSMC under control conditions as well as in the presence of a positive control (48 h 10% fetal calf serum, FCS). Figure 5B shows a slightly reduced BrdU-incorporation in EGFR-KO cells but no differences in the presence of FCS. The cell (nuclei) density was lower for EGFR-KO cells under control conditions, but increased to a similar extend as wildtype cells in the presence of FCS. Cell fraction in G2 phase was ~20% for both cell types and increased in the presence of FCS (Fig. 5A, B). Under control conditions the fraction of BrdU-positive cells in G1 and G2 was higher in WT VSMC (Fig. 5), indicating slower cell cycling of KO VSMC cells. In the presence of FCS the fractions increased substantially to values similar for both cells types. The results for EC are also shown in Fig. 5.

Next, the impact of obesity-associated stressors (48 h incubation) on BrdU-incorporation and cell cycle were assessed (Fig. 6). Metabolic or humoral stressors alone induced only slight increases in BrdU incorporation in both VSMC types. However, the combination of both stressors strongly stimulated BrdU-incorporation in WT VSMC but not in KO VSMC. BrdU-incorporation was enhanced by the combination of metabolic and humoral stressors in G1 and G2 phases (Supplementary Fig. SF11), indicating accelerated cell cycling. Consequently, cell distribution in the three phases were not affected (Supplementary Fig. SF11). Supplementary Fig. SF12

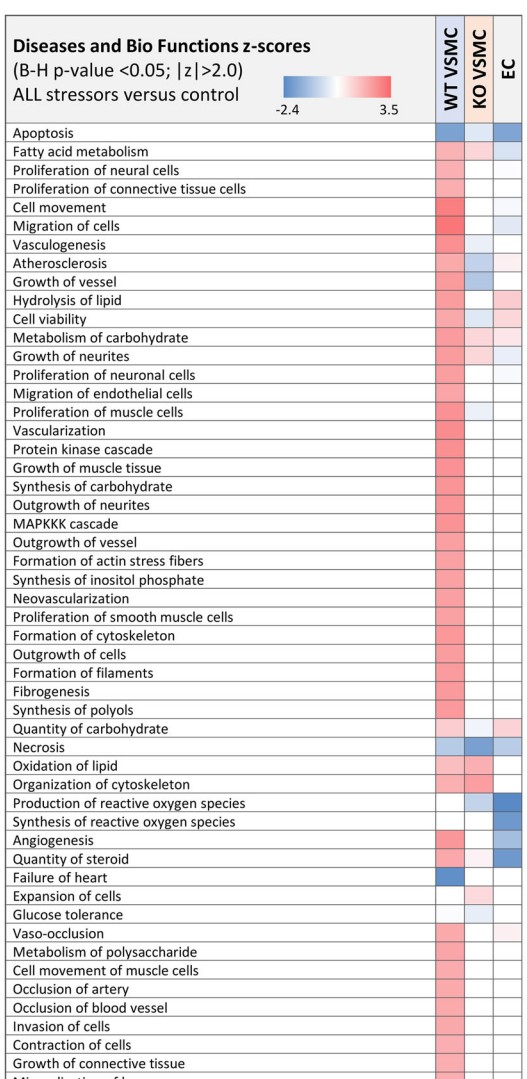

| Functions (search „vascu") | Diseases or Functions Annotation | B-H p-value | z-score |
|---|---|---|---|
| vascularization | Vascularization | 5,10E-05 | 2,91 |
| **vasculogenesis** | **Vasculogenesis** | **9,33E-14** | **2,82** |
| **angiogenesis** | **Angiogenesis** | **1,04E-15** | **2,63** |
| neovascularization | Neovascularization | 1,83E-06 | 2,40 |
| vascularization | Vascularization of eye | 1,33E-06 | 2,39 |
| migration | Migration of endothelial cells | 2,57E-05 | 2,28 |
| neovascularization | Neovascularization of eye | 2,51E-06 | 2,18 |
| **atherosclerosis** | **Atherosclerosis** | **4,34E-08** | **2,15** |
| occlusion | Occlusion of artery | 3,42E-08 | 2,15 |
| vaso-occlusion | Vaso-occlusion | 1,54E-07 | 2,15 |
| occlusion | Occlusion of blood vessel | 1,07E-07 | 2,15 |
| failure | Failure of heart | 1,76E-05 | -2,43 |

| Functions (search „cell") | Diseases or Functions Annotation | B-H p-value | z-score |
|---|---|---|---|
| **cell proliferation** | **Cell proliferation of tumor cell lines** | **7,34E-07** | **3,82** |
| cell movement | Cell movement of tumor cell lines | 2,13E-05 | 3,61 |
| migration | Migration of cells | 1,50E-13 | 3,48 |
| migration | Migration of tumor cell lines | 3,14E-05 | 3,47 |
| cell movement | Cell movement | 3,51E-13 | 3,27 |
| protein kinase cascade | Protein kinase cascade | 4,59E-04 | 2,81 |
| **proliferation** | **Proliferation of muscle cells** | **1,31E-05** | **2,78** |
| MAPKKK cascade | MAPKKK cascade | 9,64E-04 | 2,73 |
| outgrowth | Outgrowth of neurites | 2,61E-03 | 2,72 |
| development | Development of cytoplasm | 4,74E-04 | 2,62 |
| formation | Formation of cytoskeleton | 2,22E-04 | 2,62 |
| fibrogenesis | Fibrogenesis | 6,37E-04 | 2,55 |
| formation | Formation of filaments | 1,63E-03 | 2,53 |
| growth | Growth of neurites | 2,97E-04 | 2,50 |
| outgrowth | Outgrowth of cells | 7,67E-04 | 2,49 |
| **proliferation** | **Proliferation of neuronal cells** | **2,29E-04** | **2,45** |
| **proliferation** | **Proliferation of smooth muscle cells** | **4,40E-05** | **2,39** |
| formation | Formation of actin stress fibers | 3,24E-03 | 2,38 |
| migration | Migration of endothelial cells | 2,57E-05 | 2,28 |
| cell viability | Cell viability | 1,60E-05 | 2,18 |
| invasion | Invasion of cells | 1,98E-04 | 2,17 |
| invasion | Invasion of tumor cell lines | 2,86E-03 | 2,12 |
| cell movement | Cell movement of muscle cells | 6,85E-04 | 2,11 |
| **proliferation** | **Proliferation of connective tissue cells** | **1,35E-05** | **2,05** |
| **contraction** | **Contraction of cells** | **2,64E-04** | **2,03** |
| organization | Organization of cytoplasm | 6,27E-11 | 2,03 |
| organization | Organization of cytoskeleton | 1,51E-12 | 2,02 |
| proliferation | Proliferation of neural cells | 2,83E-04 | 2,01 |
| apoptosis | Apoptosis | 2,35E-16 | -2,02 |
| cell death | Cell death of tumor cell lines | 4,60E-08 | -2,24 |
| abnormal morphology | Abnormal morphology of subcellular | 3,83E-06 | -2,35 |

| Functions (search „metab") | Diseases or Functions Annotation | B-H p-value | z-score |
|---|---|---|---|
| Synthesis | Synthesis of carbohydrate | 1,09E-05 | 2,69 |
| Synthesis | Synthesis of polyols | 9,53E-04 | 2,56 |
| Metabolism | Metabolism of carbohydrate | 8,20E-08 | 2,50 |
| Hydrolysis | Hydrolysis of lipid | 2,06E-03 | 2,49 |
| synthesis | Synthesis of inositol phosphate | 2,61E-03 | 2,37 |
| metabolism | Metabolism of polysaccharide | 9,30E-04 | 2,28 |
| quantity | Quantity of steroid | 1,04E-05 | 2,18 |

**Fig. 3 | Diseases and biofunctions analysis.** IPA results for diseases and biofunctions using DEG induced by exposure to all stressors. Presented functions were filtered for z-score > 2 or < −2 and B-H corrected *p* value < 0.05. Tables to the right show details of the analysis results for WT VSMC. The complete data sets are shown in Supplementary Data 06–09.

shows the genes of the IPA terms related to proliferation and their relative expression changes in WT VSMC and KO VSMC cells. Endothelial cell cycle distribution or BrdU-incorporation were not affected by the stressors (Fig. 6 and Supplementary Fig. SF11). The data confirm the synergistic effect of the combination of stressors on RNA expression (Fig. 1A) and the predicted altered proliferation. Additionally, we observed that EGF exerted a synergistic effect with HG + FFA in WT VSMC but not in KO VSMC (Fig. 6). Furthermore, the EGFR inhibitor AG1478 prevented the effect of the combined stressors.

## Lipid accumulation

Figure 7 shows the effect of stressors on the storage of neutral lipids by the cells. Exposure of WT VSMC to metabolic stressors led to an increase of stored neutral lipids whereas exposure to humoral stressors exerted no effect. Exposure to the combination of all stressors led to a significantly larger increase in lipid storage compared to metabolic stressors alone, i.e. metabolic stressor sensitize WT VSMC for humoral stressors regarding lipid accumulation. KO VSMC did not show such a synergistic effect but only the expected effect of the metabolic stressors. Likewise, in EC lipid accumulation was enhanced by metabolic stressors but there was no synergism with humoral stressors (Fig. 7). For WT VSMC we performed an additional subanalysis regarding the subcellular localization of lipids, because we observed accumulation in the perinuclear area and as speckles in the periphery (Fig. 7). In both compartments we observed the synergistic action of the stressors. Supplementary Fig. SF13 shows the genes of the IPA terms for lipid metabolism and their relative expression changes in WT VSMC and KO VSMC cells. The data confirm the synergistic effect of the combination of stressors on RNA expression and the predicted altered lipid storage. We also assessed the involvement of EGFR in this synergistic effect. The EGFR inhibitor AG1478 prevented the effect of the combined stressors regarding lipid accumulation in WT VSMC.

Further analysis showed that the increase in cellular lipid content resulted from the accumulation of triglycerides but not of cholesterol (Fig. 7D).

## Mitochondrial function (OCR)

As free fatty acids are known to affect mitochondrial respiration, we investigated a possible synergism concerning mitochondrial function. Figure 8 shows mitochondrial function assessed by the oxygen consumption rate (OCR). Basal respiration (BR), proton leak (PL) and maximal respiration (maxOCR) were not different between WT and EGFR-KO VSMC. Metabolic stress, but not humoral stress, led to an increased proton

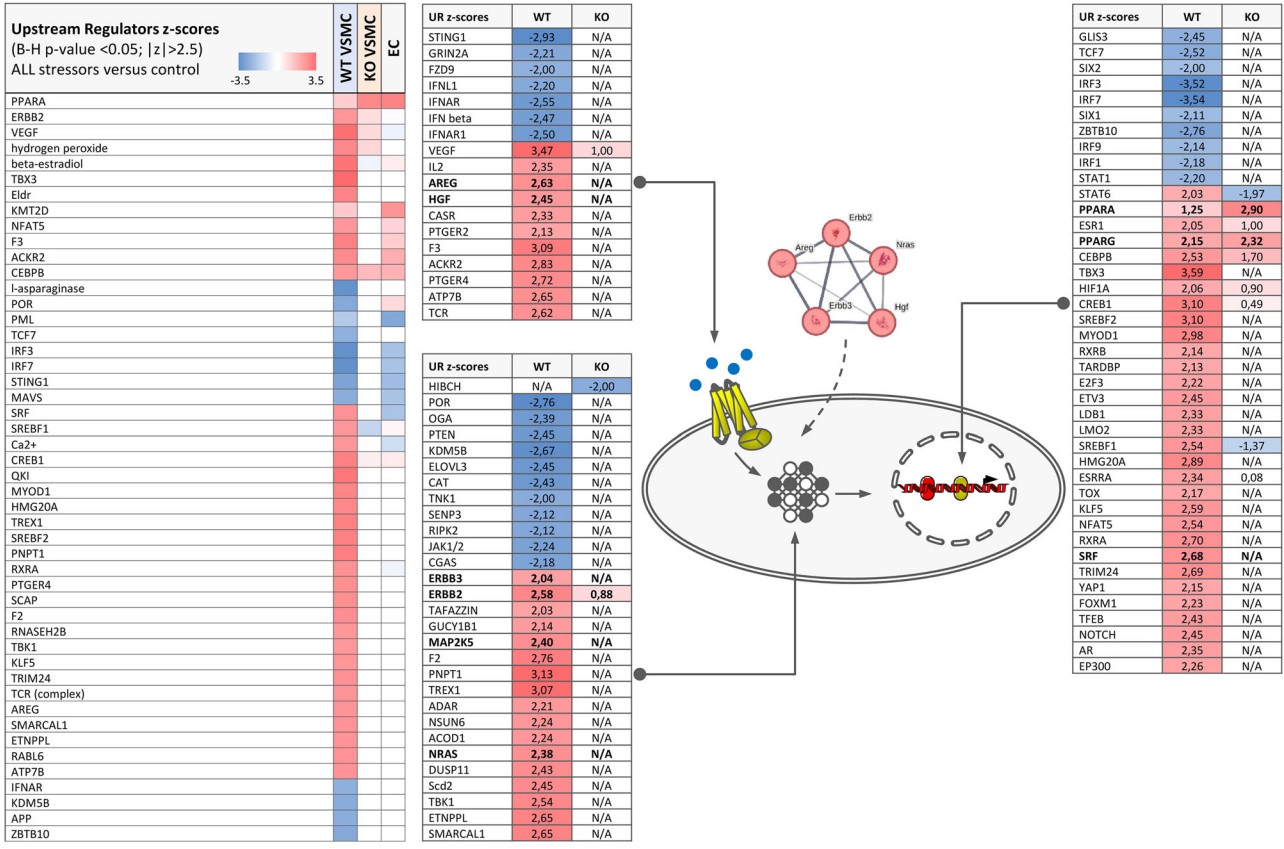

**Fig. 4 | Upstream regulator analysis.** IPA results for upstream regulators using DEG induced by exposure to all stressors. Presented functions were filtered for z-score > 2 or < −2 and B-H corrected *p* value < 0.05. Tables to the right show details of the analysis results for WT VSMC. The complete data sets are shown in Supplementary Data 10–13.

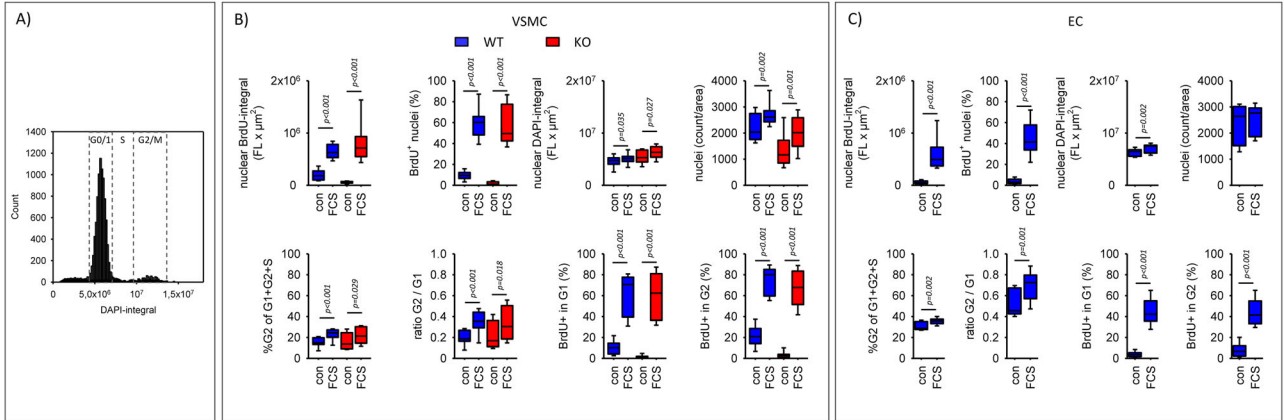

**Fig. 5 | Cell cycle analysis. A** Representative example of the nuclear DNA content ( = integral of DAPI fluorescence) distribution. **B** Baseline parameters and responsiveness to 10% FCS of WT VSMC and KO VSMC. *N* = 6 plates with 6 wells for each condition and each cell type. **C** Baseline parameters and responsiveness to 10% FCS of EC. *N* = 5 plates with at least 3 wells for each condition. Error bars above and below the box indicate the 90th and 10th percentiles.

leak in WT and EGFR-KO cells, followed by an enhanced utilization ratio (BR/maxOCR) and a reduced coupling efficiency (ATP-OCR/BR) in both cells types. There was no significant differences between the effect of metabolic stressors alone or in combination with humoral stressors.

Mitochondrial function in EC was affected in the same way as in VSMC (Fig. 8), i.e. humoral stress led to reduced coupling efficiency.

### Glucose consumption and lactate production

Supplementary Fig. SF14 shows glucose consumption and lactate production for wildtype VSMC, EGFR-KO VSMC and EC. Neither for glucose consumption nor for lactate production a synergistic effect of the stressors was observed.

### Ca²⁺-induced contractility

We assessed the $Ca^{2+}$-induced contractility of VSMC on the basis of their circularity and its changes due to a rapid increase in intracellular $Ca^{2+}$, elicited by the $Ca^{2+}$-ionophore ionomycin. When VSMC contract, the length decreases and the width increases, leading to a more circular shape and therefore to an increase in circularity. Circularity does not depend on the precise measurement of length and width, because it is derived from the

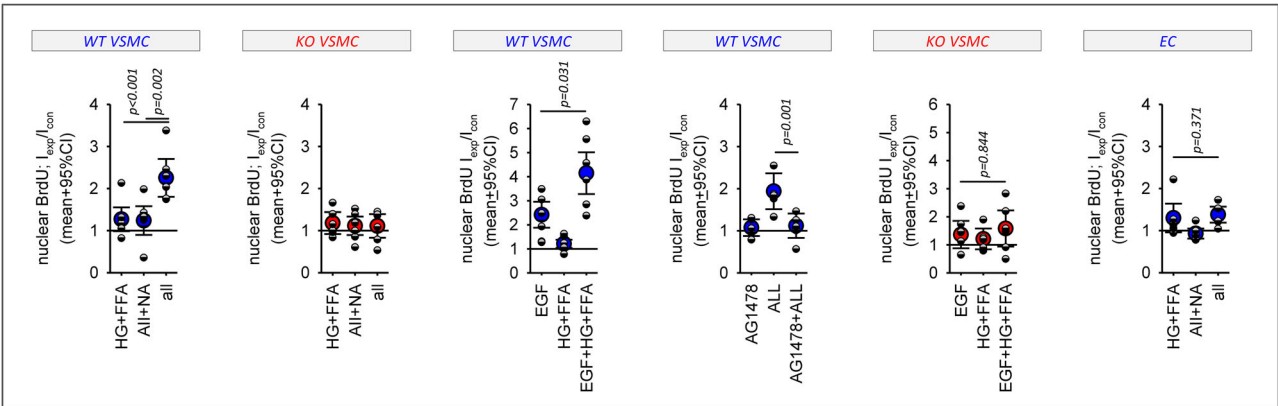

**Fig. 6 | Analysis of DNA synthesis.** Effect of the stressors on nuclear BrdU-incorporation ( = DNA synthesis) in WT VSMC, EGFR-KO VSMC and EC. In addition, the effect of EGF and of its combination with metabolic stressors on BrdU-incorporation were determined in WT VSMC and KO VSMC. Finally, the impact of the pharmaceutical EGFR-blockade (with AG1478) on the effect of all stressors was measured. $N = 6$ plates with 5 wells for each condition and each cell type. Error bars represent 95% confidence intervals.

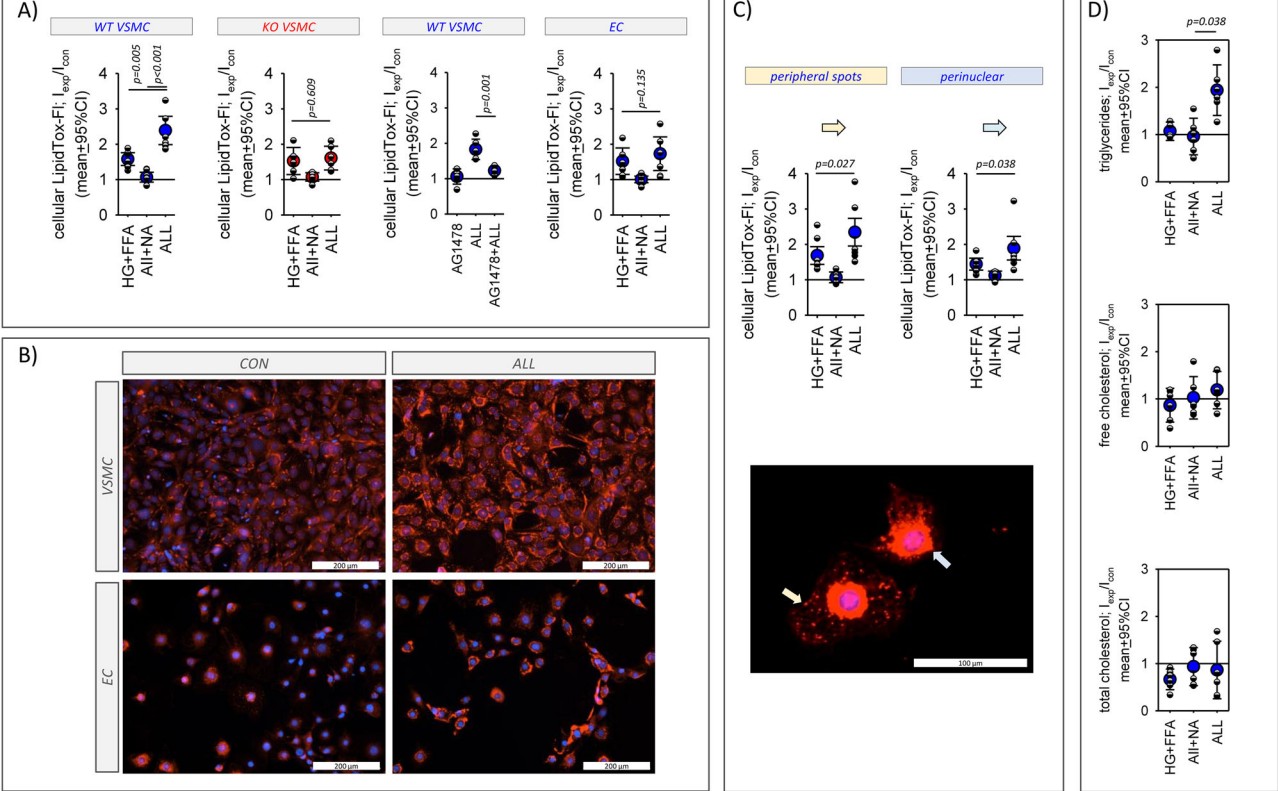

**Fig. 7 | Effect of the stressors on cellular accumulation of neutral lipids. A** Results of the stressors impact analysis for cellular lipid content in WT VSMC, KO VSMC and EC. **B** Exemplary microscopy images of VSMC and EC stained with LipidTox under control conditions and after exposure to all stressors. **C** Results of the stressors impact analysis for subcellular lipid content in WT VSMC, KO VSMC and EC. Spot-like lipid accumulation in the cell periphery (yellow arrow) and perinuclear accumulation were analyzed separately. $N = 6$ plates with 6 wells for each condition and each cell type. **D** Increased cellular lipid content resulted from triglyceride accumulation. $N = 6$. Error bars represent 95% confidence intervals.

perimeter and the area of the cell, and is therefore more robust (see methods).

Figure 9A, B show that VSMC react with a substantial acute increase in circularity in response to the addition of the $Ca^{2+}$-ionophore ionomycin. Because circularity is not normally distributed (Fig. 9A) the median circularity per sample was used as measure for the statistical analysis. In the absence of ionomycin, circularity was slightly smaller in the HG + FFA group of WT VSMC compared to the control group (Fig. 9C, D). The AII + NA and the ALL groups were not different from control. The same situation was observed in the presence of ionomycin. WT VSMC $Ca^{2+}$-induced contractility - estimated from the acute increase in circularity after the addition of ionomycin - was similar in all four groups of WT VSMC. These data are consistent with early phase events, when transdifferentiation has most probably not yet reduced contractility to a substantial extend. In EGFR-KO VSMC no differences in circularity between the four conditions were observed, neither in the absence nor in the presence of ionomycin

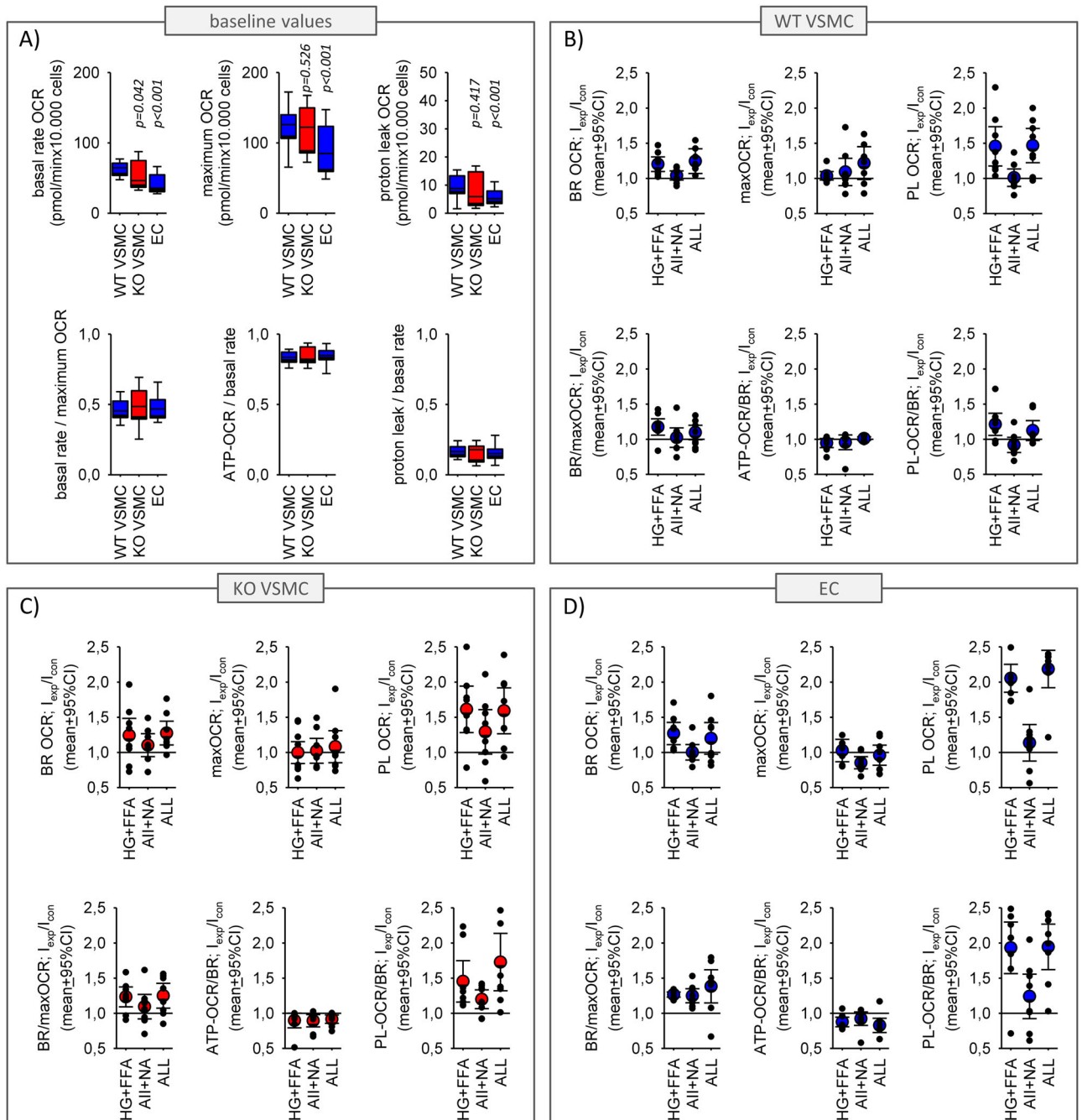

**Fig. 8 | Analysis of mitochondrial function. A** Baseline mitochondrial function in WT VSMC, KO VSMC and EC determined from the oxygen consumption rate (OCR). **B–D** Effect of the stressors on mitochondrial function. BR = basal respiration rate, PL = proton leak, ATP-OCR = ATP-synthesis dependent OCR, maxOCR = maximum OCR. *N* = 8 plates with up to 6 wells for each condition and each cell type. Error bars for scatter plots represent 95% confidence intervals. Error bars above and below the box indicate the 90th and 10th percentiles.

(Fig. 9E). In addition, we observed a reduced maximum Ca²⁺-induced contractility in KO VSMC compared to WT VSMC (Supplementary Fig. SF15). Supplementary Fig. SF15 shows the genes of the IPA terms for cell contraction and their relative expression changes in WT VSMC and KO VSMC.

**Spontaneous random motility**
Under control conditions spontaneous random motility of WT VSMC (Supplementary Fig. SF16) was in the range of published values[33]. Motility of KO VSMC was lower. Exposure to the stressors did not affect spontaneous random motility.

**Comparison of female and male VSMC**
Finally, we determined the observed synergistic impact of the stressors on lipid accumulation, BrdU-incorporation and circularity in female VSMC and compared them to the hereinabove results for male VSMC. As shown in Fig. 10 there is no synergistic action of metabolic and humoral stressors in female VSMC or EC. Thus, female WT VSMC behave similar to male KO VSMC. Supplementary Fig. SF18 shows that female WT VSMC express less EGFR compared to male VSMC. There for we analyzed some important components of downstream signaling from EGFR to the nucleus (Supplementary Fig. SF19). Our data show that female VSMC are less responsive regarding the phosphorylation of an important EGFR downstream target,

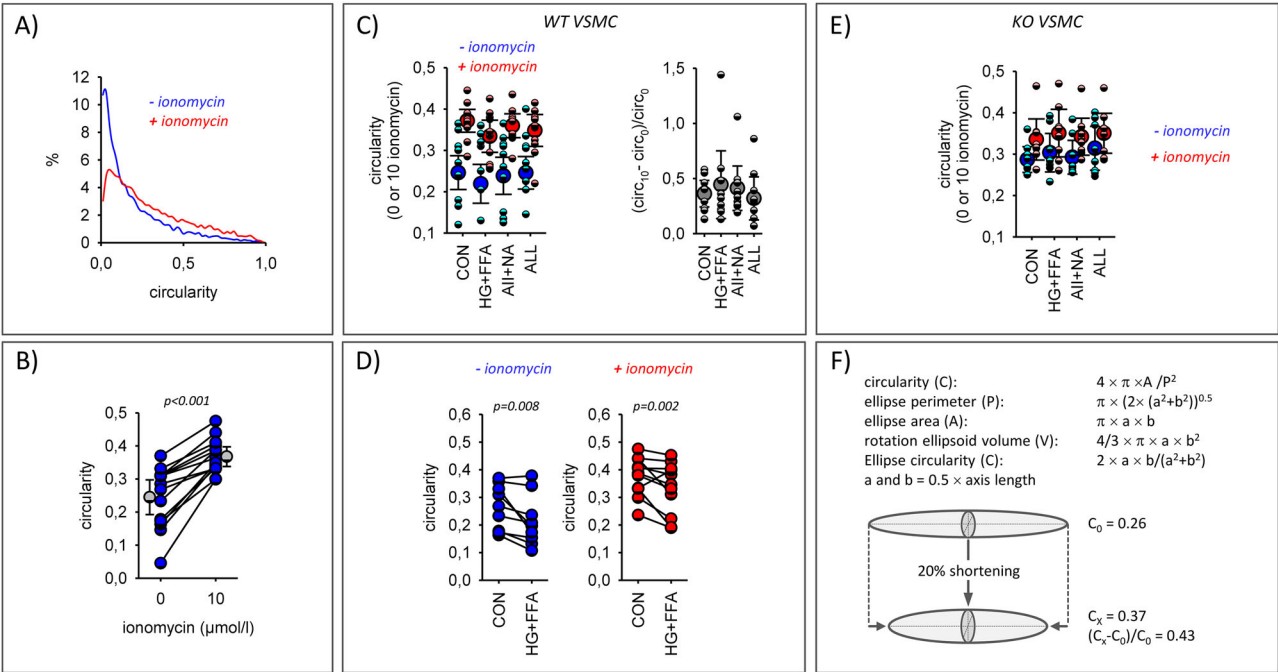

**Fig. 9 | Ca²⁺-induced cell contractility. A** Representative example VSMC circularity distribution under control conditions and after acute addition of ionomycin. **B** Increase in median VSMC circularity after acute addition of ionomycin. Results from 10 independent experiments. **C** WT VSMC circularity after exposure to the stressors before and after acute addition of ionomycin. Ca²⁺-induced contractility was estimated from the ionomycin-induced change in circularity relative to the baseline circularity. $N = 10$ plates with 6 wells for each condition. **D** Direct comparison of the cellular circularities before and immediately after addition of

ionomycin from cells exposed to control conditions or metabolic stressors. **E** KO VSMC circularity after exposure to the stressors before and after acute addition of ionomycin. $N = 8$ plates with 6 wells for each condition. **F** Modeling changes in cellular circular due acute cell shortening ( = contraction) assuming an ellipsoid shape. In this case a 20% shortening leads to a change in circularity from 0.26 to 0.37, similar to the values observed in VSMC. Error bars represent 95% confidence intervals.

**Fig. 10 | Comparing male and female cells.** Effect of the stressors on (**A**) lipid accumulation in female WT VSMC and EC ($N = 10$ and 7 plates with 6 wells for each condition and each cell type), (**B**) DNA-synthesis in female WT VSMC and EC ($N = 13$ and 8 plates with 6 wells for each condition and each cell type) and (**C**) circularity ($N = 9$–10 plates with 6 wells for each condition) in female WT VSMC. Compare with the results from male VSMC in Figs. 6, 7 and 9. Error bars represent 95% confidence intervals.

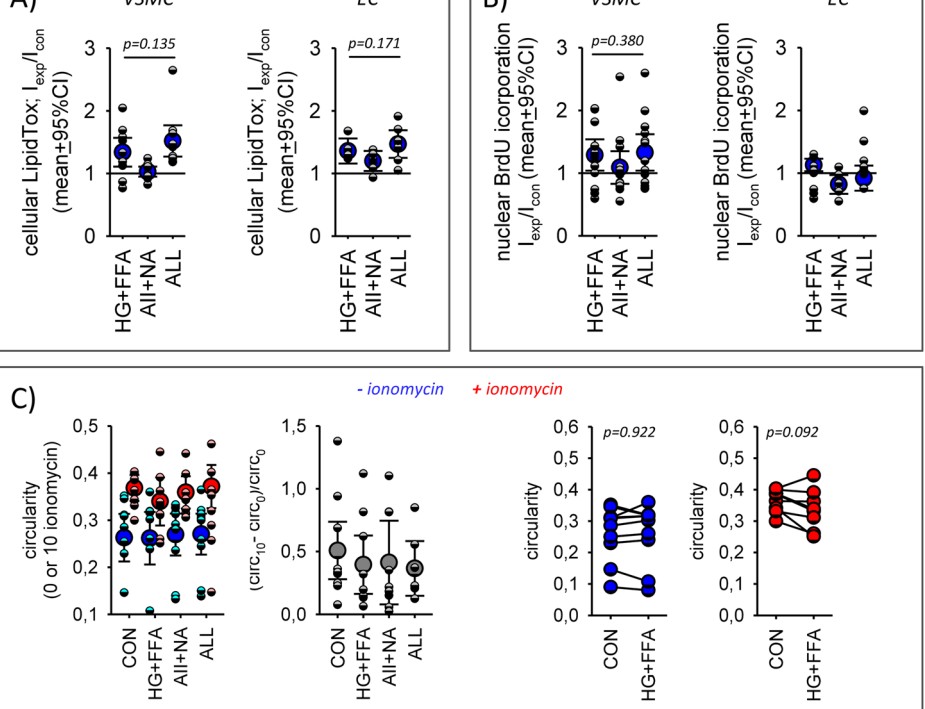

serum response factor. Incubation with stressors or EGF enhanced SRF[S103] phosphorylation in male VSMC but not in female VSMC (SF19D). By contrast, we could not detect differences in stressor effects regarding phosphorylation of ELK1 or nuclear-to-cytosol distribution of ELK1, MRTF-A or MRTF-B. However, we observed that the default distribution of ELK1 differed between male and female cells (SF19B-C). Supplementary Figs. SF14 and SF17 show that the stressors did not affect glucose handling nor exerted non-specific damage.

## Discussion

As described in the introduction, in vivo studies provided substantial evidence for a role of vascular EGFR in T2DM/obesity-associated functional and structural vascular remodeling. Data from genetic mouse models indicate that VSMC EGFR is significantly more important than endothelial EGFR[28]. These in vivo studies represented early phases of T2DM/obesity with mild vascular dysfunction and mild end-organ (kidney) damage. The in vivo data confirm the systemic relevance of VSMC-EGFR during obesity with T2DM for vascular pathophysiology but they do not allow to answer the questions concerning a direct impact of T2DM/obesity-associated stress factors on VSMC or EC.

Two important categories of T2DM/obesity-associated stress factors are (i) metabolic stressors (mainly elevated blood glucose and free fatty acid levels) and (ii) humoral stressors. Two major upregulated humoral stressors are angiotensin II and norepinephrine that represent the overactivation of the renin-angiotensin-system and the enhanced sympathetic tone, both well-known events in patients suffering from T2DM/obesity[6,7]. Therefore, we set up a study where primary vascular cells were exposed directly to metabolic or/and humoral stressors. We used murine primary WT VSMC, KO VSMC and WT EC. The stress conditions resemble the early phases of T2DM/obesity, because it is important to gain knowledge concerning initial events for a mechanistic understanding of the initial pathophysiology to support the development of preventive strategies.

Our present results, which concur with pharmacological findings[34–38], support the hypothesis that VSMC-EGFR is involved in VSMC responses to T2DM/obesity-associated stressors, because the effects were much stronger in WT VSMC compared to EGFR-KO VSMC. Furthermore, our data show that the combination of metabolic and humoral stressors exerts a synergistic effect on WT VSMC. This synergism can be easily perceptible since humoral stressors alone elicited only minor effects but enhanced the changes in the presence of metabolic stressors substantially. In other words, the humoral stressors are pathophysiological effective only in a pathological metabolic environment. In KO VSMC, this synergism was completely absent, i.e., VSMC did not respond to humoral stressors despite the pathological metabolic environment. Thus, VSMC-EGFR seems to be required for the complete impact of T2DM/obesity-associated stress factors on VSMC because it is necessary for the synergistic action of humoral and metabolic stressors. The impact of metabolic stressors alone on murine VSMC was similar between WT and KO cells, indicating that it is, at least to a substantial part, independent of EGFR. The response of EC to T2DM/obesity-associated stress factors was comparable to KO VSMC, possibly reflecting the low EGFR expression level in endothelial cells (Supplementary Fig. SF18).

After excluding non-specific cell damaging effect of the stressors (no induction of necrosis, apoptosis or cell protein loss), we started our study with a comparative analysis of transcriptome alterations in two dimensions, (i) regarding the different stressor conditions (metabolic stressors, humoral stressors, combined stressors) and (ii) regarding the different cell types (WT VSMC, KO VSMC and EC).

The initial screening for biomarker genes of phenotypic switching showed only little impact on WT VSMC and therefore no indication of major dedifferentiation. This result confirms that our setup represents early-phase conditions of the exposure to T2DM/obesity-associated stressors. Given the relative short period of exposure, this is quite appropriate. We even observed an enhanced RNA expression for TAGLN, MYLK and MYL12B, biomarkers for the contractile phenotype of VSMC. Additionally, there was an effect on two adrenergic receptors and angiotensin II receptors,

i.e., cell signaling. Of note, the observed transcriptome alterations were mainly EGFR-dependent, because only 4 of the 18 genes were also affected in EGFR-KO VSMC. Concerning a possible interaction of metabolic and humoral stressors we observed no synergism regarding these 18 genes.

Next, we performed an unbiased analysis of transcriptomic changes to predict potentially affected canonical pathways, upstream regulators of gene expression and downstream cell biological effects that might occur in the early phase of exposure to T2DM/obesity-associated stressors. At the quantitative level, there was a strong synergism of metabolic and humoral stressors concerning the number of affected genes in WT VSMC. This was true for upregulated as well as for downregulated genes. These results show that the simultaneous action of various T2DM/obesity-associated stressors leads to a strong over-additive effect that most probably represents to a certain extend the in vivo situation. Hence, it is very difficult or nearly impossible to predict the response to more than one simultaneous stimulus on the basis of their individual impacts.

Deletion of the EGFR affected the response to metabolic stressors only to a minor extend but reduced the response to humoral stressors and prevented the synergistic action completely. These results strongly indicate that VSMC-EGFR serves as an important information integration hub[39] for the different stressors at the cellular level. This observation concurs very well with in vivo data that demonstrated the important role of VSMC-EGFR for pathophysiological vascular alteration during T2DM/obesity[26]. In combination with our current data, it is conceivable that VSMC-EGFR is indeed a primary target mediating deleterious vascular effects during the initial phase of T2DM/obesity. EC responded in a similar manner as EGFR-KO VSMC.

The qualitative analysis (functional enrichment analysis, ingenuity pathway analysis) of the transcriptome data concerning canonical pathways, upstream regulators of gene expression and downstream cell biological effects predicted processes related to proliferation/cell cycle, lipid metabolism and contraction as activated in WT VSMC exposed to metabolic and humoral stressors only. Thus, these pathological relevant and T2DM/obesity-associated processes[3,6,40] were predicted to result from the synergistic, EGFR-dependent action of metabolic and humoral stressors.

Subsequently, we performed functional analyses related to these three predicted processes, using the three cell types for confirmation. Concerning proliferation, we used BrdU incorporation as a marker for DNA-synthesis. All three cell types showed a strong and comparable response to FCS as positive control. However, the responses to our stressor conditions differed. There was a clear synergistic effect of the stressors HG + FFA and AII + NA in WT VSMC from male but not from female animals and also not in KO VSMC or EC. Regarding lipid metabolism we determined lipid accumulation with a neutral lipid specific dye as a first rough approach. Of course, we are aware of lipid metabolism complexity and the necessity to address this process in more detail in future studies. Nevertheless, the results of this initial functional study show a synergistic effect of the stressors in WT VSMC from male but not from female animals and also not in KO VSMC or EC. In all three cell types metabolic stressors led to an expected increase in lipid accumulation, whereas the humoral stressors per se exerted no effect. In male WT VSMC humoral stressors intensified lipid accumulation induced by metabolic stressors in the perinuclear area as well as in the periphery significantly. This was not the case in WT VSMC from female animals, and also not in KO VSMC or EC. Possibly, the observed alterations in DNA-synthesis and lipid metabolism are the precursors of VSMC phenotypic switches that are observed in later stages of vascular damage[29,30].

The results from contractility assessment are more difficult to interpret. Basal circularity, as a rough surrogate of the contractile state, was affected slightly only by metabolic stressors in WT VSMC of male animals but not of female animals. The acute increase of cytosolic calcium, which can be assumed to be similar in both cell types with the ionomycin concentration used, resulted in a significant increase of circularity as a consequence of contraction in WT and KO VSMC, in which the response of KO VSMC was significantly smaller. This observation corresponds well to the hypotonic phenotype of the respective KO mouse models[10,12,41]. Contraction under these conditions represents the maximum $Ca^{2+}$-induced contraction and

was used to estimate the $Ca^{2+}$-dependent contractility (Δcircularity/initial circularity). None of the stressors affected contractility significantly. Thus, our results seem to not confirm the bioinformatics prediction of altered $Ca^{2+}$-dependent contractility. However, the fact that there is no clear indication for a loss of contractile characteristics is in accordance with the concept of our study, i.e. investigation of the early phase of diabetic stressor exposure, when transdifferentiation has not yet reduced contractility to a measurable extend under our experimental conditions. The same holds true for spontaneous random motility that was also not affected by the stressors. However, more detailed investigations are required in the future.

At present, our study provides an extensive phenotypic analysis and thereby substantial evidence for a sex- and EGFR-dependent stressor synergy on VSMC during T2DM. Future studies will need to further investigate specific phenotypic parameters (e.g., migration, contractility) and delve deeper into mechanistic aspects. This will include, among other things, illuminating synergy-related changes in intracellular signaling networks and identifying mechanistic reasons for the different sensitivity of female and male VSMCs. Our preliminary results indicate that in female VSMC EGFR-mediated information transfer from the surrounding milieu into the cell and nucleus (e.g. SRF) is weaker, possibly explaining in part the different responsiveness. Furthermore, the type of arterial vessel should be taken into consideration in future studies, because VSMC from conduction arteries can show a different responsiveness compared to VSMC from small resistance arteries.

Our results describe alterations of VSMC in the early phase of obesity/T2DM elicited by the synergistic impact of relevant metabolic and humoral stressors in male as well as female cells. Furthermore, we can conclude that this early phase synergistic impact depends on EGFR expression and is most probably mediated in part by EGFR-induced signaling. This study was performed with primary murine vascular cells. It is now important to compare the results obtained with murine cells to the responses of primary human vascular cells, in order to enable a translation to human pathophysiology.

## Methods
### Animals
We have complied with all relevant ethical regulations for animal use. All mouse experiments were approved by the local government (Landesverwaltungsamt Sachsen-Anhalt, Germany, Az.: 505.6.3-42502-2-1389 MLU_G; Veterinäramt Stadt Halle, Germany; Bescheid T16/2019) and conducted in accordance with the National Institutes of Health Guide for the Care and Use of Laboratory Animals, the ARRIVE guidelines and under consideration of the 3R-principle. Mice were kept at constant temperature of 22 ± 2 °C, relative humidity of 30–60%, under a 12/12 h light-dark cycle with *ad libitum* access to water and standard chow. Recently, we generated and described a conditional knock out for EGFR in VSMC via the Cre/loxP system in combination with *EGFR^{flox/flox}* C57BL/6 J mice (originally provided by Maria Sibilia, Vienna, Austria). C57BL/6 mice containing floxed *EGFR* alleles (*EGFR^{f/f}*) after removal of the *neo* cassette were used for further breeding[42]. *EGFR* was inactivated tissue-specifically in vascular smooth muscle cells (VSMC) by using *SM22-cre* transgenic mice, in which the CRE recombinase is under the control of the VSMC-specific SM22 promoter[43]. Genotyping of the mice was performed on ear punch biopsies by PCR against the floxed EGFR allele as well as CRE[10,44]. Mice were kept in the facilities of the University of Halle-Wittenberg in accordance with institutional policies and federal guidelines. Anesthesia and Euthanasia was performed with pentobarbital (400 mg per kg body weight) followed by cervical dislocation.

We are aware of the ongoing debate regarding housing conditions for animals, including housing temperature for mice on this issue (e.g.[45–48]). The thermoneutral zone for mice is above the usual and legally enforced housing temperatures. Thus, the majority of studies are performed under similar conditions as ours that are demanded by the regulatory authorities. Also, The Jackson Laboratory recommends 10-23°C (https://www.jax.org/jax-mice-and-services/customer-support/technical-support/breeding-and-husbandry-support). Since humans sojourn mainly in environments 2-3 °C below their thermoneutral zone the usual animal housing temperature seems to be adequate when biomedical questions are addressed. Furthermore, in many parts of the world free-living mice are also exposed to ambient temperatures below their thermoneutral zone, i.e. this is a natural condition. Finally, as our study was performed with isolated primary vascular cells (i.e. ex vivo) it is not directly influenced by the housing conditions.

### Murine vascular smooth muscle cell culture
Primary culture of murine VSMC were obtained from knock-out (KO) and wild-type (WT) mice, according to Ray et al.[49] and as described before[10,44,50]. Thoracic and abdominal aorta was excised in 0.9% sterile NaCl, cleared from blood as well as surrounding tissue and rinsed several times. Subsequently the aorta was transferred to DMEM-Medium ( + 10% fetal calf serum, FCS), dissected mechanically to small pieces and digested in collagenase 2 (in DMEM, 4–6 h at 37 °C, 5% $CO_2$). Thereafter, cells were gently dispersed and rinsed with medium. After 5 min centrifugation at 300xg at room temperature the cell pellet was resuspended in 5 ml fresh medium and centrifuged a second time. Finally, the cells were seeded in plastic dishes in DMEM, containing 10% FCS and incubated for 5 days at 37 °C, 5% $CO_2$. The VSMC purity achieved by our protocol was previously tested[10,44,50].

VSMC were cultivated in DMEM media (low-glucose media powder diluted in $H_2O$, with 2 g/L $NaHCO_3$, pH 7.4) supplemented with 10% FCS and 10 ng/ml bFGF. Before all experiments, the cells were synchronized by incubation in serum and supplement-free DMEM media for 24 h. The latter was also used for further incubations with obesity-associated stressors (see below). All experiments were performed on cells that underwent up to 7 passages.

### Murine endothelial cell culture
Heart and lungs were removed from the thorax and placed in a 60 mm Petri dish with Krebs-buffer (NaCl 6.9 g/l, KCl 0.35 g/l, $CaCl_2$ 0.26 g/l, $MgSO_4$ 0.296 g/l, $NaH_2PO_4$ 0.144 g/l). Hearts were carefully separated from the lungs, perfused and transferred to a separate 60 mm Petri dish with 6 ml enzyme solution (35 ml DMEM + 0.1 U/ml collagenase B + 0.8 U/ml Dispase + 20 µg/ml DNase). The organ was cut into small pieces, placed in the incubator at 37 °C / 5% $CO_2$ for 60 minutes with intermediate mixing. Subsequently the material was placed on 70 µm filter, rinsed with 8 ml DMEM-medium and centrifuged for 5 minutes at 300 g and 20°C. Cells were then suspended in 1 ml 150 mM $NH_4Cl$ + 10 mM $KHCO_3$ + 0.1 mM EDTA at pH 7.3. After 30 seconds 9 ml medium were added and the entire solution centrifuged for 5 minutes at 300 g and 20 °C. Cells were then suspended in 90 µl MCS buffer (0.5% BSA + 2 mM EDTA in PBS, pH 7.2),

20 µl CD45 magnetic beads (Miltenyi Biotec) were incubated for 15 min at 4 °C and after the addition of 1 ml MCS buffer centrifuged for 5 min at 300 g and 20 °C. A size M column was placed on the magnet, rinsed with 1 ml MCS buffer followed by the addition of the cell pellet in 500 µl MCS buffer. Columns were washed three times with 500 µl MCS buffer and the flow-through was collected (CD45⁻ cells). After centrifugation for 5 min at 300 g and 20 °C the pellet was resuspended in 90 µl MCS buffer, 20 µl CD31 magnetic beads added and the whole was incubated for 15 min at 4 °C. After addition of 1 ml MCS buffer and centrifugation for 5 minutes at $300 \times g$ and 20 °C the cell pellet (resuspended in 500 µl MCS buffer) was added to a M-size column placed on a magnet. Columns were washed 3 times with 500 µl MCS buffer. Thereafter columns were removed from the magnet, placed on a new 15 ml tube, 2 ml MCS buffer added and the cells pressed out with the plunger squeeze (CD45⁻/CD31⁺ cells). After centrifugation for 5 min at 350 g and 20 °C the cell pellet was dissolved in 1 ml of medium, placed on a fibronectin-coated 35 mm Petri dish in 1 ml of fresh DMEM medium and incubated at 5% $CO_2$ and 37°C. After reaching 80-90% confluence cells were washed with EDTA, detached with 0.5 ml of trypsin (3-5 min in the incubator) followed by stopping with 1.2 ml DMEM medium. After centrifugation for 5 min at $350 \times g$ and 20 °C the cell pellet was dissolved in 90 µl of MCS buffer and 20 µl of CD31 beads were added for a second round of clean-up (see above).

## Obesity associated stressors

Metabolic stressors consisted of high glucose concentration (20 mmol/l) and the addition of free fatty acids (100 µmol/l sodium oleat and 100 µmol/l sodium palmitate). Humoral stressors consisted of 30 nM angiotensin II and 300 nM norepinephrine (these concentration leads to maximum isometric contraction of murine aortic rings as determined in own preliminary tests[27]). To enable solubility of the free fatty acids 16.5 µM bovine serum albumin (fatty acid free) was added to all incubation solutions[26]. Stressors were applied for up to 48 h.

## RNA sample preparation

Total RNA was isolated with BluZol Reagent, following the manufacturer's instructions. To remove eventual DNA contamination traces, RNA samples were cleaned-up with NEB Monarch Total RNA mini prep kit (#T2010S), following the procedure for "TRIZOL extracted samples". RNA concentration was determined by NanoDrop (Biochrom, Germany). The quality of the to-be-sequenced RNA samples was assessed using a 2100 Bioanalyzer System (Agilent Technologies, Germany) and all samples had a RNA Integrity Number (RIN) above 7 (with 10 as maximal possible value).

## Bulk RNA sequencing

Novogene Co., Ltd (Cambridge, United-Kingdom) carried out the sequencing libraries preparation (poly(A) enrichment) and the paired-end sequencing (2 × 150 bp) runs on a NovaSeq6000 Illumina system ($N = 4$–10 for each condition, from 4–9 different male animals). Adaptor clipping and data quality control was provided by the service company as well. Read mapping to the mouse genome (mm39) was done with HISAT2[51] (v. 2.1.0) and featureCounts[52] (2.0.0, –M –t exon) was used to count the mapped reads. Gene annotation was done using BiomaRt[53] (v.2.60.0) to access Ensembl archive v109.

## Differential expression analysis

Differential expression analysis was performed using edgeR[54] (4.2.0) and DESeq2[55] (1.44.0). Due to a major influence on the data caused by the cell type, analyses were performed independently for each of them (WT VSMC, EGFR-KO VSMC, EC). For each of these analyses, based on the multiple variables influencing overall gene expression, the design with strict sample pairing of animal + treatment was applied. Genes with sufficient counts to be considered in the statistical analyses were filtered using the filterByExpr edgeR function and the independent filtering parameter ($a = 0.05$) of the DESeq2 results function. In the edgeR analysis, normalization factors were calculated with the "trimmed mean of M value" (TMM) method, and the data fitting and testing were performed with the glmGLFit and glmQLFTest functions, respectively. Significantly "differentially expressed genes" (DEG) were defined as genes with a false discovery rate (FDR) below 0.05 in both DESeq2 and edgeR outputs (overlap of the respective results), with at least 3 FPM on average in one of the sample groups considered for a given comparison and with $|\log2$ Fold Change$| \geq 0.64$ (threshold based on the inherent variation in control samples: threshold = 3 × [average coefficient of variation] from all RNAs).

## Clustering of regulated protein-coding genes

The STRING database (https://string-db.org/) and its associated tools were used to identify association networks, as well as functional clusters, among the regulated genes. For each group of genes, a full STRING network was built, with a minimum required interaction score set at 0.7 (high confidence). The edge thickness indicates the strength of the data supporting the association between two nodes, and the dotted lines represent edges between clusters. Disconnected nodes were not displayed. "K-means clustering" method served to identify clusters within the network, and enriched Gene Ontology terms and (Reactome-, KEGG-, Wiki-) pathways were automatically returned for each of them.

## Gene ontology enrichment analysis

Gene Ontology (GO) term enrichment analysis was performed with the web server g:profiler2 (https://biit.cs.ut.ee/gprofiler/orth)[31,56]. Single and multiple queries were performed. For each condition, a list of regulated genes served as input. Only GO terms comprising less than 5000 genes were considered and those were simultaneously filtered for an adjusted $p$ value below 0.05 for all datasets. The enrichment score E of the filtered GO terms was calculated, with $E$ = (intersection size/query size)/(term size/effective domain size).

## Ingenuity pathway analysis

The lists of differentially expressed genes were uploaded in the Ingenuity Pathway Analysis (IPA) application (Qiagen, Germany) to perform enrichment analysis. Core analyses were performed in order to predict which molecular and biofunctions, regulatory pathways or canonical pathways (filtered to not include overrepresented cancer-related terms) may be regulated by metabolic and humoral stressors, but also to predict which factors may be involved in the regulation of these genes (i.e. upstream regulators). Based on the IPA internal database, predicted activation states were returned for each as Z-scores (positive and negative Z-scores corresponding to putative activation and inhibition, respectively). The outputs of these analyses were aligned with the incorporated "Comparison analysis" tool. The thresholds were set at $|Z$-score$| \geq 2$ and a Benjamini Hochberg (BH) $p$ value $\leq 0.01$.

## Caspase-3 activity

Caspase-3 activity was measured as described previously[56]. Shortly, the cell culture media was removed and all wells were washed once with 1xPBS before pipetting 30 µL lysis buffer (20 mM MOPS (Merck), 0.1% Triton X-100 (Sigma) in $H_2O$, pH 7.4) per well to lyse the cells. The plates were incubated 30 min on ice. 1.5 µL of caspase substrate Ac-DEVD-AFC (AAT Bioquest) were then pipetted in each well. 30 µL MOPS-Triton and 1.5 µL Ac-DEVD-AFC were also pipetted in wells without cells (blank). The plates were incubated at 37°C and the fluorescence (excitation/emission wavelengths: 400/505 nm) was measured after 30 and 60 min with a plate reader (Infinite M200, Tecan, Germany). The caspase-3 activity development (slope between 30 and 60 min) was normalized to the protein content of the concerned well after protein concentration determination as described below.

## Necrosis measurement by Trypan blue integration

The necrotic level of the primary cells was measured using Trypan blue integration, as described before[56]. Shortly, cells were cultivated in 96-well plates and incubated for 48 h with the different substance combinations (6 wells per condition). The cell culture media was removed and all wells were incubated with 50 µL 0.2% Trypan blue (Gibco) for 5 min at 37 °C. Wells with no cells but in which Trypan blue added served to determine the blank. After washing the plates three times with 1xPBS, 100 µL 1% Triton X-100 (Sigma) was added to each well and the plates were incubated for 10 min at room temperature. The absorbance at 560 nm was measured with a plate reader (Infinite M200, Tecan, Germany). Protein concentration was determined for each well as described below so that each Trypan Blue-incorporation value was normalized to the protein content of the well.

## Protein concentration determination in 96 well-plates

Protein concentration was determined using BCA assay, as previously reported[56]. After mixing the 50 parts BCA-reagent (Thermo Scientific) with 1 part 4% $CuSO_4$, 200 µL were added to each well containing cell lysate (e.g. after caspase or necrosis measurement) or known-protein amounts (standard curve). The plates were incubated for 30 min at 37 °C, and the absorbance measured at 562 nm with a plate reader (Infinite M200, Tecan, Germany).

## Cell cycle analysis and BrdU-incorporation measurement by digital microscopy

Cells were cultivated in 96-well plates and incubated with control media or stressor-containing media (6 wells per condition). The evening before reaching 48 h incubation, Bromodeoxyuridine (BrdU) was added to each well, at a final concentration of 10 µM (24 h incubation). On the day of the measurement, the cell culture media was removed and cells were fixed with 4% Formaldehyde (Fischer). The plates were washed three times with permeabilization buffer (0.1% Triton X-100 in 1xPBS) and 2 N HCl was added to each well for 30 min at room temperature. After repeating the washing steps with permeabilization buffer, the plates were incubated for 1 h at room temperature with blocking solution (10% FCS in permeabilization buffer). The plates were incubated overnight with anti-BrdU antibodies (Becton Dickinson #347580—diluted 1:200 in 5% BSA solution). One column of wells was incubated with 5% BSA solution without antibody (blank). On the next day, the plates were washed three times with permeabilization buffer and all wells were then incubated with a red fluorescent secondary antibody (Invitrogen #A10037 - 1:250 in 5% BSA solution) for 1 h at room temperature. After repeating the washing steps, DNA was stained by incubating the plates with DAPI (Molecular Probes – final concentration: 2 µg/mL in permeabilization buffer). The plates were washed three times again and then blue- and red-fluorescence signals were acquired by digital microscopy (10× objective, Cytation3 imaging reader, BioTek, Germany). Images were analyzed with the Gene 5 Image Prime 3.16 software (BioTek, Bad Friedrichshall, Germany) and in-build routines after adjusting the necessary parameters (background, threshold, object size, rolling ball size)[57,58]. The sequence for single cell analysis was the following: 1. Identify nuclei by blue fluorescence ( = DAPI fluorescence). 2. Determine nuclear count, mean nuclear area, mean DAPI fluorescence intensity. 3. Calculate the integral of DAPI-fluorescence, as a relative marker for nuclear DNA-content, for each nucleus ( = nuclear area × mean DAPI fluorescence for each individual nucleus, after background subtraction). 4. Determine the mean nuclear red fluorescence ( = incorporated BrdU marked by AlexaFluor568-labelled antibody). 5. Calculate the integral of red fluorescence for each nucleus ( = nuclear area × mean red fluorescence for each individual nucleus, after background subtraction). 6. Identify count of BrdU-positive nuclei using a threshold value derived from the negative control. The distribution of the nuclear DAPI-integral was subsequently used to determine the relative number of nuclei in G1, S and G2 phases of the cell cycle.

## Lipid measurement

Cells were split in 96 well plates and incubated with the stressors (6 wells per condition on each plate). After the incubation period, the cells were fixed with 4% formaldehyde, washed twice with 1xPBS. 50 µL of diluted neutral lipid stain LipidTOX (Invitrogen – 1:200 in 1xPBS) supplemented with DAPI (Molecular Probes – final concentration: 2 µg/mL in 1xPBS) was added to each well and the plates were incubated for 30 min at room temperature. The blue- and red-fluorescence signals (for DNA and lipid, respectively) were acquired by digital microscopy (Cytation3 imaging reader, BioTek, Germany). Using the software Gen5 Image Prime 3.16 (BioTek, Germany), blue (nuclei) and red (cells or part of cells) objects were identified. The amount of lipids per well was estimated as follow: (mean red intensity of the red objects × area of the red objects × number of red objects) / number of blue objects. For further analysis of the type of lipid accumulated we used the Multi Lipid Detection Assay Kit (EA-7013) from Signosis (Santa Clara, CA). Cells were detached with trypsin, washed twice with cold PBS, resuspend in 1 mL of PBS and homogenized using a sonicator. Subsequently, 2 mL of chloroform and 1 mL of methanol were added to the homogenized cell sample and mixed thoroughly by vortexing for 30 s. 0.5 mL of ddH$_2$O were added to the mixture and vortexed again for 30 s to induce phase separation. Sample were then centrifuged at $1500 \times g$ for 10 min at room temperature to separate the phases. The lower chloroform phase was collected carefully and transferred to a new tube. Samples were vacuum dried until all of the chloroform was evaporated, reconstituted in PBS and assayed immediately according the manufacturer's instructions.

## Contractility assay

The cell culture media was removed after preincubation (in 96-well plate, with 6 wells per condition), and replaced by a staining solution (2 µM Calcein-AM (AAT Bioquest) and 1 µg/mL bisBenzimide H 33342 trihydrochloride (Hoechst–Sigma) diluted in Hepes-Ringer Buffer at 37 °C for 30 min, containing 1 mg/mL D-Glucose (Sigma)). The staining solution was removed and the plate was washed once with HEPES-Ringer buffer with D-Glucose. 50 µL of HEPES-Ringer buffer (37 °C) with D-Glucose were added to each well and the blue- and green-fluorescent signals (corresponding to DNA and cytoplasm staining, respectively) were acquired with a pre-heated (37 °C) Cytation3 digital microscope (BioTek, Germany). 50 µL of 20 µM ionomycin (Merck – diluted in HEPES-Ringer buffer with D-Glucose) were then added to all wells. After 5 min incubation in the pre-heated microscope, the fluorescent signals were acquired a second time. The analysis was performed using the software Gen5 Image Prime 3.08 (BioTek, Germany). Blue and green objects were identified, corresponding to cell nuclei and whole cells, respectively. The latter were filtered for objects containing only one nucleus and an area comprised between 400 µm² and 4000 µm² (to remove identified artefacts and cell clumps). Using the calculated circularity as a cell shape indicator (42), the circularity of the filtered green objects before the addition of ionomycin was used to determine the initial contractile state of the cells. On the other hand, the ratio of their circularity before and after addition of ionomycin was used to estimate the induced contraction of the smooth muscle cells. Cell circularity ($C$) was determined from cell area ($A$) and perimeter ($P$) as $C = 4 \times \pi \times A/P^2$. Theoretically, circularity can range from 0 to 1, representing perfectly linear to completely circular morphology, respectively. An acute increase in circularity results from cell contraction.

## Estimation of random motility

Random motility was estimated from the displacement over time (µm/h) of cell nuclei. For this purpose, 0.5 µg/mL bisBenzimide H 33342 trihydrochloride (Hoechst–Sigma) was added to the media for 60 min at time t = 0 h. Thereafter the cells were incubated with control media or media containing stressors (in 96-well plate, with 6 wells per condition). After 24 h the x-y-position (µm) of each nucleus in its well was determined with a Cytation3 digital microscope and using the software Gen5 Image Prime 3.08 (BioTek, Germany). X-y-positions were determined again after 3 and 5 h, and the net displacement (µm/h) calculated ( = [$\Delta x^2 + \Delta y^2$]$^{0.5}$), after appropriate mapping. Displacement rates (µm/h) between 0 and 3 h or 3 and 5 h were not different for the individual nuclei.

## Determination of oxygen consumption rate (OCR)

Oxygen consumption rate (OCR) was determined in 96-well Seahorse XF96 V3 PS Cell Culture Microplates (Agilent Technologies, Waldbronn, Germany). Real-time OCR was assessed using the Seahorse XF Cell Mito Stress Test Kit and Seahorse XFe96 Analyzer (Agilent Technologies, Waldbronn, Germany), following the manufacturer's instructions[26,59]. The mitochondrial stress test involved measuring OCR under basal conditions, followed by the sequential addition of inhibitors. Oligomycin (2 µM) was first added to inhibit complex V (ATP Synthase) of the electron transport chain (ETC). Subsequently, FCCP (4 µM) was introduced to collapse the proton gradient and disrupt the mitochondrial membrane potential. Finally, rotenone and antimycin A (0.5 µM each) were injected to inhibit complexes I and III, respectively. Hoechst 33342 (6 µM) was added during the final injection to normalize OCR to the cell number, and its induced nuclear staining was detected using digital fluorescence microscopy (Cytation3, BioTek, Bad Friedrichshall, Germany). From these measurements, the following parameters were calculated: non-mitochondrial respiration (NMOC, OCRmin_Rotenone/AntimycinA), basal respiration rate (OCRinitial value - OCRmin_Rotenone/AntimycinA), mitochondrial ATP production-linked oxygen consumption rate (OCRinitial value - OCRmin_Rotenone/AntimycinA), proton leak (OCRmin Oligomycin- OCRmin_Rotenone/AntimycinA) and coupling efficiency (((basal respiration+NMOC)-(proton leak+NMOC))/(basal respiration + NMOC)).

## Glucose consumption and lactate production

Cell culture media was used to measure glucose consumption and lactate production over the 48 h incubation. To do so, cell culture media of each well were directly transferred from the 96-well plate used for cell culture to two flat bottom 96-well plates (10 μL for lactate, or 5 μl for glucose in each plate, 6 wells per condition). For each of them, appropriate standard solutions with known concentrations, $H_2O$ and medium incubated at 37 °C for 48 h without cells, were pipetted in remaining empty wells. To the plate including glucose standard solutions, 100 μL of "Glucose-reagent" (4.4 mM ATP (Sigma), 1.6 mM NADP (Sigma), 2 U/mL Hexokinase and Glucose-6-phosphate-dehydrogenase (Sigma), in TEA Buffer) were added to each well, and the absorbance was measured at 340 nm with a plate reader (Infinite M200, Tecan, Germany) after 15 min incubation at room temperature. For the second plate, containing also lactate standard solutions, 200 μL of "Lactate-reagent" (50 μg/mL Lactate-dehydrogenase (Sigma), 3.2 mM NAD (Sigma), 0.4 M Hydrazin, 0.5 M Glycin) were added to each well, and the absorbance was measured at 340 nm with a plate reader (Infinite M200, Tecan, Germany) after 30 min incubation at 37 °C. The remaining cell culture media was removed from the cell culture plate and, after washing all wells once with 1x PBS, MOPS-Triton was used to lyse the cells. Protein concentration was determined as described hereinabove and was used as cell density estimator to normalize the obtained values. Thereby, the glucose consumption and lactate production of the cells were defined as $([Glu]_{without\ cells} - [Glu]_{treatment})$/Protein amount and $([Lactate]_{without\ cells} - [Lactate]_{treatment})$/Protein amount, respectively. Their glycolytic index was determined using the ratio of produced lactate/$2 \times$ consumed glucose.

## In-Cell fluorescence immunoassay (In-Cell FIA): Single cell immunofluorescence imaging by digital microscopy

After fixation with 4% formaldehyde for 24 h at 4 °C, cells were permeabilized (0.1% Triton X-100 in TBS; 37 mg/l Na-orthovanadate) and nonspecific antibody binding was blocked using 10% fetal calf serum in permeabilization buffer. Primary antibodies (from Cell Signaling Technologies, Frankfurt, Germany: SRF #5147, 1:2000; phospho-SRF$^{S103}$ #4261, 1:1000; ELK-1 #9182, 1:500; phospho-ELK1$^{S383}$ #9186, 1:500; MRTF-A #14760, 1:1000; MRTF-B #14613, 1:1000; EGFR #4267, 1:1000; phospho-EGFR$^{Y1068}$ #3777, 1:1000; from abcam, Cambridge, UK: HB-EGF #ab192545 1:1000) were diluted in 1% BSA in permeabilization buffer and incubated overnight at 4 °C. Anti-rabbit AlexaFluor568 secondary antibody (#A10042, Invitrogen Life Technologies, Darmstadt, Germany) was then diluted 1:500 in 1% BSA in permeabilization buffer and incubated for 1 hour in the dark at room temperature. Nuclei were stained by diluting DAPI in PBS at 1 μg/ml and applied for 10 minutes in the dark at room temperature. Digital microscopy was performed using a 40x objective. Subsequently the images were analysed with the Gene 5 3.16 software (BioTek, Bad Friedrichshall, Germany) and in-build routines after adjusting the necessary parameters (background, threshold, object size, rolling ball size). The sequence of single nucleus analysis for was the following: 1. Identify nuclei by DAPI fluorescence. 2. Determine nuclei number, mean nuclear area, mean fluorescence intensity and nuclear blue fluorescence integral. 3. Determine nuclear mean red fluorescence and nuclear red fluorescence integral ( = protein of interest marked by AlexaFluor568-labelled antibody). The sequence of nuclear-to-perinuclear ratio estimation for was the following: 1. Determination of nuclear mean red fluorescence ( = protein of interest marked by AlexaFluor568-labelled antibody) as described. 2. Define a ring area of the width = 5 μm) around each nucleus and measure the min this perinuclear compartment. 3. Calculate the ratio of mean nuclear red fluorescence and mean perinuclear red fluorescence after background subtraction.

## Immunoblotting

For protein expression level determination, cells were lysed with CST lysis buffer (20 mM Tris, pH 7.5 (Illinois Tools Works companies), 150 mM NaCl (Roth), 1% Triton X-100 (Sigma-Aldrich), 1 mM EDTA (Merck), 1 mM EGTA (Sigma-Aldrich), 184 mg/L Na-orthovanadate (Sigma-Aldrich), 2.5 mM Na-pyrophosphate (Sigma-Aldrich), 1 mM β-

glycerolphosphate (Sigma-Aldrich)), centrifuged at 13.000 g for 10 minutes and protein amount was determined with Bradford assay. Equal amounts of the proteins were denatured with 6x Laemmli buffer (0.5 M Tris pH 6.8 (Roth GmbH), 10% SDS (Roth), 10% Glycerol (Sigma-Aldrich)) at 95 °C for 5-10 minutes. Proteins were separated by 10% sodium dodecyl sulfate–polyacrylamide gel electrophoresis (SDS-PAGE) and transferred onto a nitrocellulose membrane. After blocking with 5% nonfat dry milk powder in Tris-buffered saline with Tween20 (TBS-Tween) (20 mM Tris base, pH 7.4 (Illinois Tools Works companies), 150 mM NaCl, 0.05% Tween-20 (Sigma-Aldrich)) membranes were incubated with first antibody (see below) diluted in 5% bovine serum albumin (BSA) in TBS-Tween overnight. Horse radish peroxidase (HRP)-coupled secondary antibodies, Anti-Rabbit IgG HRP, Cell Signaling #7074, 1:1000 in 5% non-fat dry milk powder in TBS-Tween) were used. After removal of unbound secondary antibody three washing steps in TBS-TWEEN were performed. Finally, Clarity™ Western ECL Substrate (Bio-Rad, Munich, Germany) was added and the peroxidase activity-based light emission was recorded by an imaging system (Image Quant LAS4000, GE Health care, Buckinghamshire, GB). The antibodies used are described below. Densitometry analysis was performed with Quantity One® software from BioRad (Feldkirchen, Germany) and the relative expression values calculated under consideration of membrane protein staining signal (Ponceau) or GAPDH expression.

## Statistics and Reproducibility

Data are presented either as box plots or as mean ± 95% confidence intervals. ANOVA on rank test followed by post hoc testing or Wilcoxon rank sum tests were used because pre-test data analysis by SigmaPlot 12.5 indicated that most of the data were not normally distributed. Biometrical planning was performed under consideration of the 3R-principle with $\alpha = 0.05$ and $\beta = 0.8$. For cell culture experiments, cells from at least 4 different animals were used. Supplementary Fig. SF1 shows the detailed numbers. Data from all experiments that proceeded technically according to plan were included into the analyses.

Our experimental design provided strictly connected sets of samples for control, metabolic stressors, humoral stressors and combined stressors originating from the same animal, at the same cell passage, treated at exactly the same time (Supplementary Fig. SF1). Thus, each set was biologically independent. This design allowed us to calculate the relative effects of the stressors in a paired way (stressor effect$_i$ = value$_{i,stressors}$/value$_{i,control}$ for $i \in [1 - N]$, with $N$ the final number of replicates), which greatly reduced effects of passage or animal. Subsequently, we could calculate the 95% or 99% confidence intervals and test the exclusion of the value = 1 (corresponding to a difference from controls with $\alpha < 0.05$ or 0.01, respectively). This procedure was applied in part to the analysis of gene expression as well as to the functional assays. The final number of independent biological replicates is given in each figure legend or in the text.

## Ethics approval and consent to participate

We have complied with all relevant ethical regulations for animal use. All mouse experiments were approved by the local government (Landesverwaltungsamt Sachsen-Anhalt, Germany, Az.: 505.6.3-42502-2-1389 MLU_G; Veterinäramt Stadt Halle, Germany; Bescheid T16/2019) and conducted in accordance with the National Institutes of Health Guide for the Care and Use of Laboratory Animals, the ARRIVE guidelines and under consideration of the 3R-principle.

## Reporting summary

Further information on research design is available in the Nature Portfolio Reporting Summary linked to this article.

## Material availability

The RNASeq datasets generated during and/or analysed during the current study are available in the **gene expression omnibus database** with the study identity GSE294868 (https://www.ncbi.nlm.nih.gov/geo/query/acc.cgi?acc=GSE294868). All other data sets are provided in the supplementary

data files of this publication. Numerical source data are provided in Supplementary Data 24.

## Data availability
The RNASeq datasets generated during and/or analysed during the current study are available in the *gene expression omnibus database* with the study identity GSE294868 (https://www.ncbi.nlm.nih.gov/geo/query/acc.cgi?acc=GSE294868). All other data sets are provided in the Supplementary Data files of this publication. Numerical source data are provided in Supplementary Data 24.

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

## Acknowledgements
The Seahorse experiments were performed at the core facility of the Medical Faculty, Martin Luther University Halle-Wittenberg, Halle, Germany. This study was funded by the Deutsche Forschungsgemeinschaft (DFG GE905/24-1 and DFG SCH1450/2-4).

## Author contributions
All authors read and approved the final manuscript. V.D.: Substantial contribution to acquisition of data, analysis and interpretation of data, RNASeq data processing and analysis, bioinformatic analysis, revising the article critically for important intellectual content, final approval of the version to be published. S.R.: Substantial contribution to acquisition of data, analysis of data, revising the article critically for important intellectual content, final approval of the version to be published. N. N-A.: Substantial contributions to design, acquisition of data, analysis and interpretation of data, final approval of the version to be published. B.S.: Substantial contributions to conception and design, acquisition of data, analysis and interpretation of data, drafting the article and revising it critically for important intellectual content, final approval of the version to be published. M.K.: Substantial contribution to acquisition of data, revising the article critically for important intellectual content, final approval of the version to be published. S.M.: Substantial contribution to acquisition of data, analysis and interpretation of data, revising the article critically for important intellectual content, final approval of the version to be published. G.S.: Substantial contributions to design, analysis and interpretation of data, revising the article critically for important intellectual content, final approval of the version to be published. M.G.: Substantial contributions to conception and design, acquisition of data, analysis and interpretation of data, drafting the article and revising it critically for important intellectual content, final approval of the version to be published.

## Funding

## Competing interests
The authors declare no competing interests.

## Consent for publication
All authors read and approved the final manuscript and gave their consent for publication.

## Additional information
**Supplementary information** The online version contains Supplementary Material available at https://doi.org/10.1038/s42003-025-09416-7.

