## [Transparent Peer Review file · Communications Biology]

Early-phase impact of obesity-associated stress on murine vascular smooth muscle cells depends on EGFR and sex

Corresponding Author: Professor Michael Gekle

Version 0:

Reviewer comments:

Reviewer #1

(Remarks to the Author)

This study investigated the impact of obesity-associated metabolic and humoral stress on primary VSMC and endothelial cells from mice with conditional EGFR knockout (KO) and wildtype (WT) animals, focusing on early-phase impact and synergistic effects.

The study has been well done and the authors have done an excellent job on the multidisciplinary approaches used. However there are some concerns that warrant further consideration.

Specific comments

- 1) A strength of the study is that the cells studied were derived directly from vessels (primary culture) rather than immortalized or commercial cells. However studies were performed in cells passaged to passage 7. Passaging itself is associated with phenotypic changes and this needs to be considered. In particular, it is well known that VSMCs that are passaged undergo changes in expression of the Ang II receptor- hence it would be essential to show AT1R expression in the KO and WT cells at low and higher passaged cells.
- 2) While the molecular phenotypic is comprehensive, the functional studies are less specific. It would be important to use direct parameters of cell growth, migration, contraction, de-differentiation etc. For examples, changes in Ca²⁺ signaling are not directly indices of contraction.
- 3) One of the most interesting findings relates to the sexual dimorphism. However there are no attempts to unravel the underlying mechanisms for this important observation.
- 4) A weakness of the study is that it is entirely in vitro- If in vivo studies cant be performed, ex vivo studies should at least be included- eg vascular function in isolated vessels.
- 5) Studies were performed in VSMCs and ECs from large arteries. Many vascular changes in obesity are evident in small arteries. Ideally it would be important to study cells from small vessels too – or at least include this in th discussion.
- 6) This is an in vitro study that has comprehensively phenotyped VSMCs and endothelial cells exposed to various metabolic and vasoactive stimuli. It is entirely descriptive and would be strengthened by some underlying mechanistic insights

Reviewer #2

(Remarks to the Author)

General comments:

This manuscript describes experiments characterizing alteration in vascular smooth muscle cells (VSMC) in the early stages of obesity/T2DM in mice. It was observed that obesity-associated metabolic and humoral stressors induce synergistic transcriptomic effects in male WT-VSMC, initiating proliferative and lipogenic dedifferentiation. This early-phase effect

requires EGFR and was not observed in female WT-VSMC or in male EC.

The study is very complete, and the results are interesting. This reviewer has no major concerns.

Specific comments:

Abstract: Please state a hypothesis (Same in Introduction).

Line 112: I know it is common practice to house mice at $22\pm 2^{\circ}\text{C}$. But this is not their thermal neutral zone. Please consider the following publication:

AW Fischer, B Cannon, J Nedergaard. Optimal housing temperatures for mice to mimic thermal environment of humans: An experimental study. *Mol Metab* 7:161-170, 2017.

Version 1:

Reviewer comments:

Reviewer #1

(Remarks to the Author)

The authors have partially addressed my concerns, although most of the responses are defensive.

There are still concerns

I still believe that there is a need to show the biological functional responses beyond the molecular phenotyping.

A comprehensive study such as this should be able to evaluate functional responses such as cell growth (proliferation/hypertrophy/apoptosis), cell migration etc.

I appreciate the efforts to address the interesting sex differences. However the studies should not be 'preliminary' - Definitive experiments with enough power should be included in this paper.

Dear reviewers, dear editorial office,

The following table lists all the comments of the reviewers together with our responses and the changes made. We are thankful for the valuable advises and hope that our revised manuscript meets the criteria for publication.

Comment	Response										
Reviewer 1 A strength of the study is that the cells studied were derived directly from vessels (primary culture) rather than immortalized or commercial cells. However studies were performed in cells passaged to passage 7. Passaging itself is associated with phenotypic changes and this needs to be considered. In particular, it is well known that VSMCs that are passaged undergo changes in expression of the Ang II receptor- hence it would be essential to show AT1R expression in the KO and WT cells at low and higher passaged cells.	We are aware of the fact that the expression of At1r declines rapidly in primary culture. Also, for that reason we applied a strictly paired approach where always one control sample and one sample for each stressor condition were generated and analysed in parallel, thereby avoiding an unequal impact of confounders. Thus, the decline in At1r affects all incubation conditions in the same way. Unfortunately, there are no antibodies against At1r of sufficient quality available (at least under our experimental conditions), making expression analysis at the protein level impossible. Alternatively, we compared the RNA expression of At1ar (obtained by RNASeq) for different passages. As expected, we observed a drop in expression from passage 2 to 3, whereas the level remained stable thereafter:   <caption>Approximate data from the FPM At1ar plot</caption>   Passage Mean FPM At1ar     #2 ~32   #3 ~11   #4 ~12   #7 ~11   	Passage	Mean FPM At1ar	#2	~32	#3	~11	#4	~12	#7	~11
Passage	Mean FPM At1ar										
#2	~32										
#3	~11										
#4	~12										
#7	~11										
While the molecular phenotypic is comprehensive, the functional studies are less specific. It would be important to use direct parameters of cell growth, migration, contraction, de-differentiation etc. For examples, changes in Ca²⁺ signaling are not directly indices of contraction.	We agree that more direct functional parameters should be addressed in future studies and we are planning to do so. Yet this was beyond the scope of the present study. Direct cell growth assessment for primary vascular cells is difficult since their growth rate is very low as long as they retain some state of differentiation. Thus, more direct parameters of cell growth are difficult to obtain within 48h. Concerning migration we obtained, as also mentioned in the text, only a first rough impression on spontaneous single cell motility. Future studies will have to refine migration assessment, including bulk migration and directed migration [DOI: 10.1016/j.bbamcr.2016.03.017]. As a parameter for contractility, we determined acute calcium-induced cell shortening by determination of cell										

	circularity. In our opinion this is a valid parameter to gain information regarding this function in primary cell culture. We now mention these limitations at the end of the discussion. [lines 747-754: “At present, our study provides an extensive phenotypic analysis and thereby substantial evidence for a sex- and EGFR-dependent stressor synergy on VSMC during T2DM. Future studies will need to further investigate specific phenotypic parameters (e.g., migration, contractility) and delve deeper into mechanistic aspects. This will include, among other things, illuminating synergy-related changes in intracellular signaling networks and identifying mechanistic reasons for the different sensitivity of female and male VSMCs. Our preliminary results indicate that in female VSMC EGFR-mediated information transfer from the surrounding milieu into the cell and nucleus (e.g. SRF) is weaker, possibly explaining in part the different responsiveness.”].
One of the most interesting findings relates to the sexual dimorphism. However there are no attempts to unravel the underlying mechanisms for this important observation.	The different expression levels of EGFR in female and male VSMC (supplementary figure 18) is a first hint to a possible mechanism explaining differences. We included some additional preliminary data obtained in the meantime in the supplements and mention this issue in the results [lines 638-641: “Preliminary data (supplementary figure SF19) in addition show that female VSMC are less responsive regarding the phosphorylation of an important EGFR downstream target, serum response factor. Incubation with stressors or EGF enhanced SRF^{S103} phosphorylation in male VSMC but not in female VSMC.”] as well as in the discussion [lines 747-754: “At present, our study provides an extensive phenotypic analysis and thereby substantial evidence for a sex- and EGFR-dependent stressor synergy on VSMC during T2DM. Future studies will need to further investigate specific phenotypic parameters (e.g., migration, contractility) and delve deeper into mechanistic aspects. This will include, among other things, illuminating synergy-related changes in intracellular signaling networks and identifying mechanistic reasons for the different sensitivity of female and male VSMCs. Our preliminary results indicate that in female VSMC EGFR-mediated information transfer from the surrounding milieu into the cell and nucleus (e.g.

	SRF) is weaker, possibly explaining in part the different responsiveness.”]. Further, more in depth investigations will follow in future studies.
A weakness of the study is that it is entirely in vitro- If in vivo studies cant be performed, ex vivo studies should at least be included- eg vascular function in isolated vessels.	T2DM-associated alterations of vascular function and the role of EGFR had been studied before. These studies included in vivo and ex vivo experiment that showed (i) the pathological impact of T2DM and a contribution of EGFR, especially VSMC-EGFR [DOI: 10.1007/s00125-020-05187-4; DOI: 10.1038/s41598-021-86587-3; DOI: 10.3390/biomedicines1108224]. The aim of the present study was the more direct cellular analysis under controlled conditions. This is not possible in an animal model, where e.g. long term feeding would be required and a multitude of parameters are affected. Ex vivo exposure over 48h of vessels is, according to our experience, not feasible.
Studies were performed in VSMCs and ECs from large arteries. Many vascular changes in obesity are evident in small arteries. Ideally it would be important to study cells from small vessels too – or at least include this in the discussion.	We included this aspect at the end of the discussion. [lines 747-756: “At present, our study provides an extensive phenotypic analysis and thereby substantial evidence for a sex- and EGFR-dependent stressor synergy on VSMC during T2DM. Future studies will need to further investigate specific phenotypic parameters (e.g., migration, contractility) and delve deeper into mechanistic aspects. This will include, among other things, illuminating synergy-related changes in intracellular signaling networks and identifying mechanistic reasons for the different sensitivity of female and male VSMCs. Our preliminary results indicate that in female VSMC EGFR-mediated information transfer from the surrounding milieu into the cell and nucleus (e.g. SRF) is weaker, possibly explaining in part the different responsiveness. Furthermore, the type of arterial vessel should be taken into consideration in future studies, because VSMC from conduction arteries can show a different responsiveness compared to VSMC from small resistance arteries.”].
This is an in vitro study that has comprehensively phenotyped VSMCs and endothelial cells exposed to various metabolic and vasoactive stimuli. It is entirely descriptive and would be strengthened by some underlying mechanistic insights	We agree, that in future, after completing the comprehensive and necessary phenotyping at the transcriptional and cellular level, more mechanistic studies will have to follow. However, it was necessary to generate a solid basis, as done in this study. Some initial insights were already obtained regarding EGFR-SRF signaling which is of eminent importance for VSMC

	[http://dx.doi.org/10.1016/j.molcel.2016.10.016; DOI: 10.1152/physrev.00041.2003]. We included some additional data in the supplements and mention this issue in the results [lines 638-641: “Preliminary data (supplementary figure SF19) in addition show that female VSMC are less responsive regarding the phosphorylation of an important EGFR downstream target, serum response factor. Incubation with stressors or EGF enhanced SRF^{S103} phosphorylation in male VSMC but not in female VSMC.”] as well as in the discussion [lines 747-754: “At present, our study provides an extensive phenotypic analysis and thereby substantial evidence for a sex- and EGFR-dependent stressor synergy on VSMC during T2DM. Future studies will need to further investigate specific phenotypic parameters (e.g., migration, contractility) and delve deeper into mechanistic aspects. This will include, among other things, illuminating synergy-related changes in intracellular signaling networks and identifying mechanistic reasons for the different sensitivity of female and male VSMCs. Our preliminary results indicate that in female VSMC EGFR-mediated information transfer from the surrounding milieu into the cell and nucleus (e.g. SRF) is weaker, possibly explaining in part the different responsiveness.”].
Reviewer 2	
Abstract: Please state a hypothesis (Same in Introduction).	A hypothesis has been added.
Line 112: I know it is common practice to house mice at 22±2°C. But this is not their thermal neutral zone. Please consider the following publication: AW Fischer, B Cannon, J Nedergaard. Optimal housing temperatures for mice to mimic thermal environment of humans: An experimental study. Mol Metab 7:161-170, 2017.	Thank you for this comment. The conditions of our animal husbandry have to follow strictly the guidelines of the state regulatory department and are as follows: The animal husbandry rooms have a HEPA filtered air supply (HOSCH Filter EU13) and a directional air flow. Floors, ceilings and walls have a plastic coating and are resistant to disinfectants and hydrogen peroxide. The joints are welded and sealed (gas-tight construction). Only individually ventilated cage systems (IVC cages type GreenLine, Sealsafe Plus, Tecniplast) are used as animal husbandry systems. The climate parameters (light periodicity, humidity, room temperature, air exchange) are centrally regulated, monitored and comply with the legal requirements (45 — 65% relative humidity, 20-24°C, 15-fold air exchange).

COMMSBIO-25-6067-T (R2)

List of changes:

In accordance with the requests and specifications of the editorial office the following changes were made. We are thankful for the valuable advises and hope that our revised manuscript meets the criteria for publication.

Page 2, lines 41-43: The following sentence has been added:

“Male WT-VSMC showed higher EGFR-expression than female WT-VSMC and responded with enhanced SRF^{S103}-phosphorylation, a classical downstream target of EGFR, to the stressors.”

Page 5, lines 122-133: The following paragraph has been added:

We are aware of the ongoing debate regarding housing conditions for animals, including housing temperature for mice on this issue (e.g.^{32, 33, 34, 35}). The thermoneutral zone for mice is above the usual and legally enforced housing temperatures. Thus, the majority of studies are performed under similar conditions as ours that are demanded by the regulatory authorities. Also, The Jackson Laboratory recommends 10-23°C (<https://www.jax.org/jax-mice-and-services/customer-support/technical-support/breeding-and-husbandry-support>). Since humans sojourn mainly in environments 2-3 °C below their thermoneutral zone the usual animal housing temperature seems to adequate when biomedical questions are addressed. Furthermore, in many parts of the world free-living mice are also exposed to ambient temperatures below their thermoneutral zone, i.e. this is a natural condition. Finally, as our study was performed with isolated primary vascular cells (i.e. ex vivo) it is not directly influenced by the housing conditions.

Page 9, lines 297-305: The following paragraph regarding additional lipid determination has been added:

For further analysis of the type of lipid accumulated we used the Multi Lipid Detection Assay Kit (EA-7013) from Signosis (Santa Clara, CA). Cells were detached with trypsin, washed twice with cold PBS, resuspend in 1 mL of PBS and homogenized using a sonicator. Subsequently, 2 mL of chloroform and 1 mL of methanol were added to the homogenized cell sample and mixed thoroughly by vortexing for 30 seconds. 0.5 mL of ddH₂O were added to the mixture and vortexed again for 30 seconds to induce phase separation. Sample were then centrifuged at 1,500 x g for 10 minutes at room temperature to separate the phases. The lower chloroform phase was collected carefully and transferred to a new tube. Samples were vacuum dried until all of the chloroform was evaporated, reconstituted in PBS and assayed immediately according the manufacturer's instructions.

Page 10-11, lines 371-393: The following paragraph regarding additional In-Cell fluorescence immunoassay has been added:

In-Cell fluorescence immunoassay (In-Cell FIA): Single cell immunofluorescence imaging by digital microscopy

After fixation with 4% formaldehyde for 24 hours at 4°C, cells were permeabilized (0.1% Triton X-100 in TBS; 37 mg/l Na-orthovanadate) and non-specific antibody binding was blocked using 10% fetal calf serum in permeabilization buffer. Primary antibodies (from Cell Signaling Technologies, Frankfurt, Germany: SRF #5147, 1:2000; phospho-SRF^{S103} #4261, 1:1000; ELK-1 #9182, 1:500; phospho-ELK1^{S383} #9186, 1:500; MRTF-A #14760, 1:1000; MRTF-B #14613, 1:1000; EGFR #4267, 1:1000; phospho-EGFR^{Y1068} #3777, 1:1000; from abcam, Cambridge, UK: HB-EGF

#ab192545 1:1000) were diluted in 1% BSA in permeabilization buffer and incubated overnight at 4°C. Anti-rabbit AlexaFluor568 secondary antibody (#A10042, Invitrogen Life Technologies, Darmstadt, Germany) was then diluted 1:500 in 1% BSA in permeabilization buffer and incubated for 1 hour in the dark at room temperature. Nuclei were stained by diluting DAPI in PBS at 1µg/ml and applied for 10 minutes in the dark at room temperature. Digital microscopy was performed using a 40x objective. Subsequently the images were analysed with the Gene 5 3.16 software (BioTek, Bad Friedrichshall, Germany) and in-build routines after adjusting the necessary parameters (background, threshold, object size, rolling ball size). The sequence of single nucleus analysis for was the following: 1. Identify nuclei by DAPI fluorescence. 2. Determine nuclei number, mean nuclear area, mean fluorescence intensity and nuclear blue fluorescence integral. 3. Determine nuclear mean red fluorescence and nuclear red fluorescence integral (= protein of interest marked by AlexaFluor568-labelled antibody). The sequence of nuclear-to-perinuclear ratio estimation for was the following: 1. Determination of nuclear mean red fluorescence (= protein of interest marked by AlexaFluor568-labelled antibody) as described. 2. Define a ring area of the width = 5 µm) around each nucleus and measure the min this perinuclear compartment. 3. Calculate the ratio of mean nuclear red fluorescence and mean perinuclear red fluorescence after background subtraction.

Page 20-21, lines 632-641: The following sentence together with figure 7D regarding additional lipid determination has been added:

Further analysis showed that the increase in cellular lipid content resulted from the accumulation of triglycerides but of cholesterol (Fig. 7D).

Page 24, lines 703-711: The following sentences have been added:

Thus, female WT VSMC behave similar to male KO VSMC. Supplementary Figure SF18 shows that female WT VSMC express less EGFR compared to male VSMC. There for we analyzed some important components of downstream signaling from EGFR to the nucleus (supplementary figure SF19). Our data show that female VSMC are less responsive regarding the phosphorylation of an important EGFR downstream target, serum response factor. Incubation with stressors or EGF enhanced SRF^{S103} phosphorylation in male VSMC but not in female VSMC (SF19D). By contrast, we could not detect differences in stressor effects regarding phosphorylation of ELK1 or nuclear-to-cytosol distribution of ELK1, MRTF-A or MRTF-B. However, we observed that the default distribution of ELK1 differed between male and female cells (SF19B-C).

Supplementary figure SF19 has been added to the supplementary material.